# Mitigating Covariate Shift in Imitation Learning via Offline Data Without Great Coverage

**Jonathan D. Chang**[*]
Department of Computer Science
Cornell University
jdc396@cornell.edu

**Masatoshi Uehara**[*]
Department of Computer Science
Cornell University
mu223@cornell.edu

**Dhruv Sreenivas**
Department of Computer Science
Cornell University
ds844@cornell.edu

**Rahul Kidambi**[†]
Amazon Search & AI
rk773@cornell.edu

**Wen Sun**
Department of Computer Science
Cornell University
ws455@cornell.edu

## Abstract

This paper studies offline Imitation Learning (IL) where an agent learns to imitate an expert demonstrator without additional online environment interactions. Instead, the learner is presented with a static offline dataset of state-action-next state transition triples from a potentially less proficient behavior policy. We introduce Model-based IL from Offline data (`MILO`): an algorithmic framework that utilizes the static dataset to solve the offline IL problem efficiently both in theory and in practice. In theory, even if the behavior policy is highly sub-optimal compared to the expert, we show that as long as the data from the behavior policy provides sufficient coverage on the expert state-action traces (and with no necessity for a global coverage over the entire state-action space), `MILO` can provably combat the covariate shift issue in IL. Complementing our theory results, we also demonstrate that a practical implementation of our approach mitigates covariate shift on benchmark MuJoCo continuous control tasks. We demonstrate that with behavior policies whose performances are less than half of that of the expert, `MILO` still successfully imitates with an extremely low number of expert state-action pairs while traditional offline IL methods such as behavior cloning (BC) fail completely. Source code is provided at https://github.com/jdchang1/milo.

## 1 Introduction

*Covariate shift* is a core issue in Imitation Learning (IL). Traditional IL methods like behavior cloning (BC) [49], while simple, suffer from covariate shift, learning a policy that can make arbitrary mistakes in parts of the state space not covered by the expert dataset. This leads to compounding errors in the agent's performance [57], hurting the generalization capabilities in practice.

Prior works have presented several means to combat this phenomenon in IL. One line of thought utilizes an *interactive expert*, i.e. an expert that can be queried at an arbitrary state encountered during the training procedure. Interactive IL algorithms such as DAgger [59], LOLS [15], DART [40], and AggreVaTe(D) [58; 66] utilize a reduction to no-regret online learning and demonstrate that under certain conditions, they can successfully learn a policy that imitates the expert. These interactive IL algorithms, however, cannot provably avoid covariate shift if the expert is not *recoverable*. That is,

---

[*]Equal contribution

[†]Work done outside Amazon

35th Conference on Neural Information Processing Systems (NeurIPS 2021), virtual.

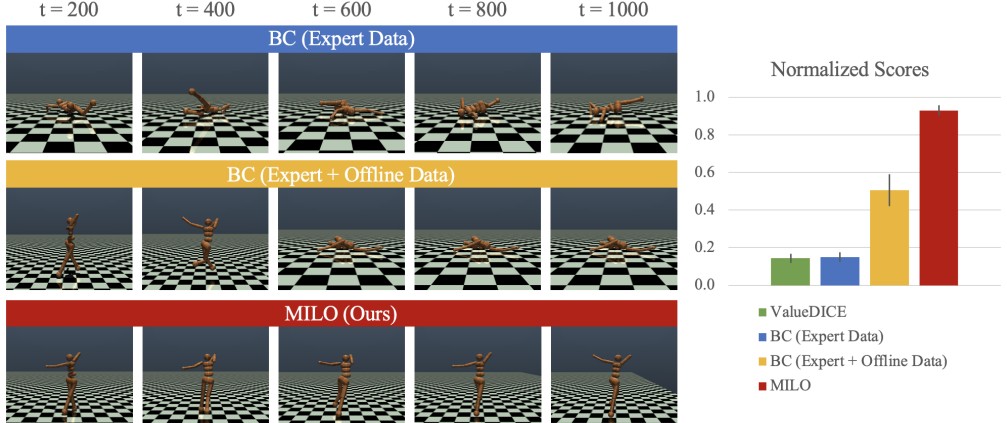

Figure 1: (Left) Frames at timesteps 200, 400, 600, 800, and 1000 for `Humanoid-v2` from policies trained with BC on 100 state-action pairs from the expert (blue), BC on 1M offline samples plus 100 expert samples (yellow), and our algorithm `MILO` (red). The expert has a performance of 3248 and the behavior policy used to collect the offline dataset has performance of $1505 \pm 473$ ($\approx 46\%$ of the expert's). (Right) Expert performance normalized scores averaged across 5 seeds.

$A^{\pi_e}(s, a) = \Theta(H)$ where $\pi_e$ is the expert, $A^\pi$ is the usual (dis)advantage function,[3] and $H$ is the planning horizon [52; 4, Chapter 15]. A second line of work that avoids covariate shift utilizes either a known transition dynamics model [2; 86] or uses real world interactions [27; 10; 65; 38; 56; 33]. Prior works have shown that with known transition dynamics or real world interactions, agents can provably avoid covariate shift in both tabular and general MDPs [4; 52] even without a recoverable expert. While these results offer strong theoretical guarantees and empirical performance, online interactions are often costly and prohibitive for real world applications where active trial-and-error exploration in the environment could be unsafe or impossible. A third perspective towards addressing this issue is to assume that the expert visits *the entire state space* [62], where the expert effectively informs the learner what actions to take in every state. Unfortunately, such a full coverage expert distribution might be rare and holds only for special MDPs and expert policies (for e.g. an expert that induces ergodicity in the MDP).

In this work, we consider a new perspective towards handling the covariate shift issue in IL. In particular, we investigate a pure offline learning setting where the learner has access to neither the expert nor the environment for additional interactions. The learner, instead, has access to a small pre-collected dataset of state-action pairs sampled from the expert and a large batch offline dataset of state-action-next state transition triples sampled from a behavior policy that could be highly sub-optimal (see Figure 1 where BC on the offline data results in a low-quality policy). Unlike prior works that require online interactions, our proposed method, `MILO` performs high fidelity imitation in an offline, data-driven manner. Moreover, different from interactive IL, we do not require the expert to be present during learning, significantly relieving the expert's burden. Finally, in contrast to the prior work [62] that assumes the expert distribution covers the entire state-action space (i.e., $\max_\pi \max_{s,a} d^\pi(s, a)/d^{\pi^e}(s, a) < \infty$ where $d^\pi$ denotes the state-action distribution of policy $\pi$), we require the offline dataset to provide *partial coverage*, i.e., it only needs to cover the expert's state-actions (i.e., $\max_{s,a} d^{\pi^e}(s, a)/\rho(s, a) < \infty$ where $\rho$ is the offline distribution of some behavior policy).[4]

In summary, we list our main contributions below:

1. We propose Model based Imitation Learning from Offline data, `MILO`: a model-based framework that leverages offline batch data with only partial coverage (see Section 4.1 for definition) to overcome covariate shift in IL.

---

[3]in this work, we use cost instead of reward, thus we call $A^\pi$ the disadvantage function.

[4]In our analysis, we refine the density ratio $d^{\pi^e}(s, a)/\rho(s, a)$ via the concept of relative conditional number which allows us to extend it to large MDPs where the ratio is infinite but the relative condition number is finite.

2. Our analysis is modular and covers common models such as discrete MDPs and linear models. Notably, our new result on non-parametric models (e.g. Gaussian Processes) with relative condition number is new even considering all existing results in offline RL (see Remark 4,7,11).

3. The practical instantiation of our general framework leverages neural network model ensembles, and demonstrates its efficacy on benchmark MuJoCo continuous control problems. Specifically, even under low-quality behavior policies, our approach can successfully imitate using an extremely small number of expert samples while algorithms like BC completely fail (Figure 1).

## 1.1 Related work

**Imitation Learning** As summarized above, avoiding covariate shift in IL is an important topic. Another relevant line of research is IL algorithms that use *offline* or *off-policy* learning. ValueDICE [38] presents a principled way to leverage off-policy data for IL. In theory, the techniques from ValueDICE (and more broadly, DICE [46; 84]) require the data provided to the agent to have global coverage. Moreover in practice, ValueDICE uses online interaction and maintains an increasing replay buffer which may eventually provide global coverage. Instead, we aim to study offline IL without any online interactions and are interested in the setting where offline data does not have global coverage. Another line of work [29; 14] studies IL in an offline setting by only using the expert dataset. In contrast to these works, our goal is to study the use of an additional offline dataset collected from a behavior policy to mitigate covariate shift, as information theoretically any algorithm that relies solely on expert data will still suffer from covariate shift in the worst case [52].

Similar in setting, Cascaded Supervised IRL (CSI) [35] performs imitation offline with both an expert dataset and a static offline dataset by first fitting a reward function that is then used in Least Squares Policy Iteration. CSI, however, requires the expert data to have global coverage and does not mitigate covariate shift with partial coverage like MILO does. Finally, Variational Model-Based Adversarial IL (V-MAIL) [50] learns a dynamics model from a static offline dataset and performs offline imitation within the model. V-MAIL, however, studies zero-shot IL where the static offline data is samples from a variety of source tasks, and the expert dataset is samples from the transfer target task. In contrast, MILO investigates avoiding covariate shift in IL with an offline dataset collected by a potentially suboptimal policy from the same task.

**Offline RL** In offline RL, algorithms such as FQI [22] have finite-sample error guarantees under the global coverage [45; 5]. Recently, many algorithms to tackle this problem have been proposed from both model-free [76; 68; 43; 26; 23; 39] and model-based perspectives [82; 34; 44] with some pessimism ideas. The idea of pessimism features in offline RL with an eye to penalize the learner from visiting unknown regions of the state-action space [54; 31; 79; 12]. We utilize pessimism within the IL context where, unlike RL, the learner does not have access to an underlying reward signal. Our work expands prior theoretical results by (a) formalizing the partial coverage condition using a notion of a relative condition number, and (b) offering distribution-dependent results when working with non-parametric models including gaussian processes. See Appendix C for a detailed literature review.

## 2 Setting

We consider an episodic finite-horizon Markov Decision Process (MDP), $\mathcal{M} = \{\mathcal{S}, \mathcal{A}, P, H, c, d_0\}$, where $\mathcal{S}$ is the state space, $\mathcal{A}$ is the action space, $P : \mathcal{S} \times \mathcal{A} \to \Delta(\mathcal{S})$ is the MDP's transition, $H$ is the horizon, $d_0$ is an initial distribution, and $c : \mathcal{S} \times \mathcal{A} \to [0, 1]$ is the cost function. A policy $\pi : \mathcal{S} \to \Delta(\mathcal{A})$ maps from state to distribution over actions. We denote $d_P^\pi \in \Delta(\mathcal{S} \times \mathcal{A})$ as the average state-action distribution of $\pi$ under transition kernel $P$, that is, $d_P^\pi = 1/H \sum_{t=1}^{H} d_{P,t}^\pi$, where $d_{P,t}^\pi \in \Delta(\mathcal{S} \times \mathcal{A})$ is the distribution of $(s^{(t)}, a^{(t)})$ under $\pi$ at $t$. Given a cost function $f : \mathcal{S} \times \mathcal{A} \mapsto [0, 1]$, $V_{P,f}^\pi$ denotes the expected cumulative cost of $\pi$ under the transition kernel $P$ and cost function $f$. Following a standard IL setting, the ground truth cost function $c$ is unknown. Instead, we have the demonstrations by the expert specified by $\pi_e : \mathcal{S} \to \Delta(\mathcal{A})$ (potentially stochastic and not necessarily optimal). Concretely, we have an expert dataset in the form of i.i.d tuples $\mathcal{D}_e = \{s_i, a_i\}_{i=1}^{n_e}$ sampled from distribution $d_P^{\pi_e}$.

In our setting, we also have an offline static dataset consisting of i.i.d tuples $\mathcal{D}_o = \{s_i, a_i, s_i'\}_{i=1}^{n_o}$ s.t. $(s, a) \sim \rho(s, a), s' \sim P(s, a)$, where $\rho \in \Delta(\mathcal{S} \times \mathcal{A})$ *is an offline distribution* resulting from some behavior policies. Note behavior policy could be a much worse policy than the expert $\pi^e$. Our goal is

---

**Algorithm 1** Framework for model-based Imitation Learning with offline data (`MILO`)

---

1: **Require**: IPM class $\mathcal{F}$, model class $\mathcal{P}$, policy class $\Pi$, datasets $\mathcal{D}_e = \{s, a\}$, $\mathcal{D}_o := \{s, a, s'\}$
2: Train Dynamics Model and Bonus: $\widehat{P} : \mathcal{S} \times \mathcal{A} \to \mathcal{S}$ and $b : \mathcal{S} \times \mathcal{A} \to \mathbb{R}^+$ on offline data $\mathcal{D}_o$
3: Pessimistic model-based min-max IL: with $\widehat{P}, b, \mathcal{D}_e$, obtain $\hat{\pi}_{\mathrm{IL}}$ by solving the following:

$$\hat{\pi}_{\mathrm{IL}} = \mathrm{argmin}_{\pi \in \Pi} \max_{f \in \mathcal{F}} \left[ \mathbb{E}_{(s,a) \sim d_{\hat{P}}^\pi} [f(s,a) + b(s,a)] - \mathbb{E}_{(s,a) \sim \mathcal{D}_e}[f(s,a)] \right] \quad (1)$$

---

to only leverage $(\mathcal{D}_e + \mathcal{D}_o)$ to learn a policy $\pi$ that performs as well as $\pi_e$ with regard to optimizing the ground truth cost $c$. More specifically, our goal is to utilize the offline static data $\mathcal{D}_o$ to combat covariate shift and learn a policy that can significantly outperform traditional offline IL methods such as Behavior cloning (BC), *without any interaction with the real world or expert*.

**Function classes** We introduce function approximation. Since we do not know the true cost function $c$ and transition kernel $P$, we introduce a cost function class $\mathcal{F} \subset \mathcal{S} \times \mathcal{A} \to [0, 1]$ and a transition model class $\mathcal{P} : \mathcal{S} \times \mathcal{A} \to \Delta(\mathcal{S})$. We also need a policy class $\Pi$. For the analysis, we assume realizability:

**Assumption 1.** $c \in \mathcal{F}, P \in \mathcal{P}, \pi_e \in \Pi$.

We use Integral Probability Metric (IPM) as a distribution distance measure, i.e., given two distributions $\rho_1$ and $\rho_2$, IPM with $\mathcal{F}$ is defined as $\max_{f \in \mathcal{F}} \left[ \mathbb{E}_{(s,a) \sim \rho_1}[f(s,a)] - \mathbb{E}_{(s,a) \sim \rho_2}[f(s,a)] \right]$.

## 3 Algorithm

The core idea of `MILO` is to imitate the expert by optimizing an IPM distance between the agent and the expert with a penalty term for pessimism over the policy class. `MILO` consists of three steps:

1. **Model learning:** fit a model $\hat{P}$ from the offline data $\mathcal{D}_o$ to learn $P$,

2. **Pessimistic penalty design:** construct penalty function $b(s, a)$ such that there is a high penalty on state-action pairs that are not covered by the offline data distribution $\rho$.

3. **Offline min-max model-based policy optimization:** optimize Eq. (1)

Algorithm 1 provides the details of `MILO`. We explain each component in detail as follows.

**Model learning and Penalty:** Our framework assumes we can learn a calibrated model $(\hat{P}, \sigma)$ from the dataset $\mathcal{D}_o$, in the sense that for any $s, a$, we have: $\left\| \hat{P}(\cdot|s,a) - P(\cdot|s,a) \right\|_1 \leq \min\{2, \sigma(s,a)\}$. Such model training is possible in many settings including classic discrete MDPs, linear models (KNR [32]), and non-parametric models such as GP. In practice, it is also common to train a model ensemble based on the idea of bootstrapping and then use the model-disagreement to approximate $\sigma$. With such a calibrated model, the penalty will simply be $b(s, a) = O(H\sigma(s,a))$. We will formalize this model learning assumption in Section 4. We give several examples below.

*For any discrete MDP*, we use the empirical distribution, i.e., $\hat{P}(s'|s,a) = N(s', s, a)/(N(s,a) + \lambda)$, where $N(s, a)$ is the number of $(s, a)$ in $\mathcal{D}_o$, and $N(s', s, a)$ is the number of $(s, a, s')$ in $\mathcal{D}_o$, and $\lambda \in \mathbb{R}^+$. In this case, we can set $\sigma(s, a) = \widetilde{O}\left( \sqrt{|\mathcal{S}|/N(s,a)} \right)$. See example 1 for more details.

*For continuous Kernelized Nolinear Regulator* (KNR [32]) model where the ground truth transition $P(s'|s,a)$ is defined as $s' = W^\star \phi(s,a) + \epsilon$, $\epsilon \sim \mathcal{N}(0, \Sigma)$, with $\phi$ being a (nonlinear) feature mapping, we can learn $\widehat{P}$ by classic Ridge regression on offline dataset $\mathcal{D}_o$. Here we can set $\sigma(s, a) = \widetilde{O}\left( \beta \sqrt{\phi(s,a)^\top \Sigma_{n_o}^{-1} \phi(s,a)} \right)$ for some $\beta \in \mathbb{R}^+$, where $\Sigma_o$ is the data covariance matrix $\Sigma_{n_o} := \sum_{i=1}^{n_o} \phi(s_i, a_i) \phi(s_i, a_i)^\top + \lambda \mathbf{I}$. See example 2 for more details.

*For non-parametric nonlinear model such as Gaussian Process (GP)*, under the assumption that $P$ is in the form of $s' = g^\star(s,a) + \epsilon$, $\epsilon \sim \mathcal{N}(0, \Sigma)$ (here $\mathcal{S} \subset \mathbb{R}^{d_\mathcal{S}}$), we can simply represent $\hat{P}$ using GP posteriors induced by $\mathcal{D}_o$, i.e., letting GP posterior be $GP(\hat{g}, k_{n_o})$, we have $\hat{P}(s'|s,a)$ being

represented as $s' = \hat{g}(s,a) + \epsilon$. Then, we can set $\sigma(s,a) = \widetilde{O}\left(\beta k_{n_o}\left((s,a),(s,a)\right)\right)$ with some parameter $\beta \in \mathbb{R}^+$ (see example 3 for more details). GP is a powerful model and has been being widely used in robotics problems, see [36; 18; 8; 71; 25] for examples.

In practice, we can also use a model ensemble of neural networks with the maximum disagreement between models as $\sigma$. This has been widely used in practice (e.g., [47; 6; 48]). We leave the details to Section 5 where we instantiate a practical version of MILO, and the experiment section. As we can see from the examples mentioned above, in general, the penalty $b(s,a) = O(H\sigma(s,a))$ is designed such that it has a high value in state-action space that is not covered well by the offline data $\mathcal{D}_o$, and has a low value in space that is covered by $\mathcal{D}_o$. Adding such a penalty automatically forces our policy to stay away from these regions where $\hat{P}$ is not accurate. On the other hand, for regions where $\rho$ has good coverage (thus $\hat{P}$ is accurate), we force $\pi$ to stay close to $\pi^e$.

**Pessimistic model-based min-max IL:** Note Eq. 1 is purely computational, i.e., we do not need any real world samples. To solve such min-max objective, we can iteratively (1) perform the best response on the max player, i.e., compute the $\arg\max$ discriminator $f$ given the current $\pi$, and (2) perform incremental update on the min player, e.g., use policy gradient (PG) methods (e.g. TRPO) inside the learned model $\hat{P}$ with cost function $f(s,a) + b(s,a)$. We again leave the details to Section 5.

### 3.1 Specialization to offline RL

In RL, the cost function $c$ is given. The goal is to obtain $\pi^* = \arg\max_{\pi \in \Pi} V^{\pi}_{P,c}$. The pessimistic policy optimization procedure [81; 31] is $\hat{\pi}_{\mathrm{RL}} = \arg\min_{\pi \in \Pi} \mathbb{E}_{(s,a) \sim d^{\pi}_{\hat{P}}}[c(s,a) + b(s,a)]$. While this is not our main contribution, we will show a byproduct of our result is a novel non-parametric analysis for offline RL which does not assume $\rho$ has global coverage (see Remarks 4,7,11).

## 4   Analysis

Our algorithm depends on the model $\hat{P}$ estimated from the offline data. We provide a unified analysis assuming that $\hat{P}$ is calibrated in that its confidence interval is provided. Specifically, we assume:

**Assumption 2.** *With probability* $1 - \delta$, *the estimate model* $\hat{P}$ *satisfies the following:* $\|\hat{P}(\cdot|s,a) - P(\cdot|s,a)\|_1 \le \min(\sigma(s,a),2) \quad \forall(s,a) \in \mathcal{S} \times \mathcal{A}$. *We set the penalty as* $b(s,a) = H\min(\sigma(s,a),2)$.

We give the following three examples. For details, refer to appendix A.

**Example 1** (Discrete MDPs). *Set uncertainty measure* $\sigma(s,a) = \sqrt{\frac{|\mathcal{S}|\log 2 + \log(2|\mathcal{S}||\mathcal{A}|/\delta)}{2\{N(s,a)+\lambda\}}} + \frac{\lambda}{N(s,a)+\lambda}$.

**Example 2** (KNRs). *In KNRs, the ground truth model is* $s' = W^*\phi(s,a) + \epsilon$, $\epsilon \sim \mathcal{N}(0,\zeta^2\mathbf{I})$, *where* $s \in \mathbb{R}^{d_{\mathcal{S}}}, a \in \mathbb{R}^{d_{\mathcal{A}}}$, $\phi : \mathcal{S} \times \mathcal{A} \mapsto \mathbb{R}^d$ *is some known state-action feature mapping. The estimator is*

$$\hat{g}(\cdot) = \hat{W}\phi(\cdot), \quad \hat{W} = \arg\min_{W \in \mathbb{R}^{d_{\mathcal{S}} \times d_{\mathcal{A}}}} 1/n_o \sum_{(s,a) \in \mathcal{D}_o}[\|W\phi(s,a) - s'\|_2^2] + \lambda\|W\|_F^2,$$

*where* $\|\cdot\|_F$ *is a frobenius norm. We set the uncertainty measure* $\sigma(s,a)$:

$$\sigma(s,a) = (1/\zeta)\beta_{n_o}\sqrt{\phi^{\top}(s,a)\Sigma^{-1}_{n_o}\phi(s,a)}, \quad \Sigma_{n_o} = \sum_{i=1}^{n_o}\phi(s_i,a_i)\phi^{\top}(s_i,a_i) + \lambda\mathbf{I}$$

*with* $\beta_{n_o} = \{2\lambda\|W^*\|_2^2 + 8\zeta^2[d_{\mathcal{S}}\log(5) + \log(1/\delta) + \bar{\mathcal{I}}_{n_o})]\}^{1/2}$, *where* $\bar{\mathcal{I}}_{n_o} = \log(\det(\Sigma_{n_o}/\lambda\mathbf{I}))$.

**Example 3** (GPs). *In GPs, the ground truth model is defined as* $s' = g^*(s,a) + \epsilon$, $\epsilon \sim \mathcal{N}(0,\zeta^2\mathbf{I})$ *where* $g^*$ *belongs to an RKHS* $\mathcal{H}_k$ *with a kernel* $k(\cdot,\cdot)$. *Denote* $x := (s,a)$, *we have GP posterior as*

$$\hat{g}(\cdot) = S(\mathbf{K}_{n_o} + \zeta^2\mathbf{I})^{-1}\bar{k}_{n_o}(\cdot), \quad S = [s'_1, \cdots, s'_{n_o}] \in \mathbb{R}^{d_{\mathcal{S}} \times n_o}, \quad \bar{k}_{n_o}(x) = [k(x_1,x), \cdots, k(x_{n_o},x)]^{\top},$$

$$\{\mathbf{K}_{n_o}\}_{i,j} = k(x_i,x_j)\,(1 \le i \le n_o, 1 \le j \le n_o), \quad k_{n_o}(x,x') = k(x,x') - \bar{k}_{n_o}(x)^{\top}(\mathbf{K}_{n_o} + \zeta^2\mathbf{I})^{-1}\bar{k}_{n_o}(x'),$$

*with* $\sigma(\cdot) = \beta_{n_o}k_{n_o}(\cdot,\cdot)/\zeta$, $\beta_{n_o} = O((d_{\mathcal{S}}\log^3(d_{\mathcal{S}}n_o/\delta)\mathcal{I}_{n_o})^{1/2})$, $\mathcal{I}_{n_o} = \log(\det(\mathbf{I} + \zeta^{-2}\mathbf{K}_{n_o}))$.

**General results**   We show our general error bound results. For the proof, refer to appendix B. For analytical simplicity, we assume $|\mathcal{F}|$ is finite (but the bound only depends on $\ln(|\mathcal{F}|))$ [5].

---

[5] When $|\mathcal{F}|$ is infinite, we can show that the resulting error bound scales w.r.t its metric entropy.

**Theorem 3** (Bound of `MILO`). *Suppose assumptions 1,2. Then, with probability $1 - 2\delta$,*

$$V_{P,c}^{\hat{\pi}_{\text{IL}}} - V_{P,c}^{\pi_e} \leq \text{Err}_o + \text{Err}_e, \ \text{Err}_o = 8H^2 \mathbb{E}_{(s,a) \sim d_P^{\pi_e}}[\min(\sigma(s,a), 1)], \ \text{Err}_e = 2H\sqrt{\frac{\log(2|\mathcal{F}|/\delta)}{2n_e}}.$$

We will show through a set of examples where $\mathbb{E}_{(s,a) \sim d_P^{\pi_e}}[\min(\sigma(s,a), 1)]$ shrinks to zero as $n_o \to \infty$ under the partial coverage, i.e., when $\rho$ covers $d_P^{\pi_e}$. Asymptotically, $\text{Err}_e$ will dominate the bound. Note that $\text{Err}_e$ has two components, a linear $H$ and a term that corresponds to the statistical error related to expert samples and function class complexity. Comparing to BC, which has a rate $O(H^2\sqrt{\log(|\Pi|)/n_e})$ [4, Chapter 14], we see that the horizon dependence is improved.

Before going to each analysis of $\text{Err}_o$, we highlight two important points in our analysis. First, our bound requires only the partial coverage, i,e., it depends on $\pi_e$-concentrability coefficient which measures the discrepancy between the offline data and expert data. This is the first work deriving the bound with $\pi_e$-concentrability coefficient in IL with offline data. Second, our analysis covers non-parametric models. This is a significant contribution as previous pessimistic offline RL finite-sample error results have been limited to the finite-dimensional linear models or discrete MDPs [31; 54].

**Remark 4** (Implications on offline RL). *As in theorem 3, we have $V_{P,c}^{\hat{\pi}_{\text{RL}}} - V_{P,c}^{\pi^*} = O(H^2 \mathbb{E}_{(s,a) \sim d_P^{\pi^*}}[\sigma(s,a)])$ (appendix B). Note similar results have been obtained in [82; 34]. Since this term is $\text{Err}_o$ by just replacing $\pi_e$ with $\pi^*$, this offline RL result is a by-product of our analysis.*

### 4.1 Analysis: Discrete MDPs

We start from discrete MDP as a warm up. Denote $C^{\pi_e} = \max_{(s,a)} d_P^{\pi_e}(s,a)/\rho(s,a)$.

**Theorem 5.** *Suppose $\lambda = \Omega(1)$ and the partial coverage $C^{\pi_e} < \infty$. With probability $1 - \delta$,*

$$\text{Err}_o \leq c_1 H^2 \left( \sqrt{\frac{C^{\pi_e}|\mathcal{S}|^2|\mathcal{A}|}{n_o}} + \frac{C^{\pi_e}|\mathcal{S}||\mathcal{A}|}{n_o} \right) \cdot \log(|\mathcal{S}||\mathcal{A}|c_2/\delta),$$

*where $c_1, c_2$ are universal constants.*

The error does not depend on $\sup_{\pi \in \Pi} C^{\pi}$ or $\bar{C} = \sup_{\pi \in \Pi} \max_{(s,a)} d_P^{\pi}(s,a)/d_P^{\pi_e}(s,a)$. We only require the *partial coverage* $C^{\pi_e} < \infty$, which is much weaker than $\sup_{\pi \in \Pi} C^{\pi} < \infty$ ($\rho$ has global coverage) and $\bar{C} < \infty$ ($d_P^{\pi_e}$ has global coverage [62]). When $C^{\pi_e}$ is small and $n_o$ is large enough, $\text{Err}_e = O\left(H\sqrt{|\mathcal{S}||\mathcal{A}|/n_e}\right)$ dominates $\text{Err}_o$ in theorem 3. Then, the error is linear in horizon $H$.

### 4.2 Analysis: KNRs and GPs for Continuous MDPs

Now we move to continuous state-action MDPs. In continuous MDPs, assuming the boundedness of density ratio $C^{\pi_e}$ is still a strong assumption. As we dive into the KNR and the nonparametric GP model, we will replace the density ratio with a more refined concept *relative condition number*.

**KNRs** Let $\Sigma_\rho = \mathbb{E}_{(s,a) \sim \rho}[\phi(s,a)\phi(s,a)^\top]$ and $\Sigma_{\pi_e} = \mathbb{E}_{(s,a) \sim d_P^{\pi_e}}[\phi(s,a)\phi(s,a)^\top]$. We define the *relative condition number* as $C^{\pi_e} = \sup_{x \in \mathbb{R}^d} \left( \frac{x^\top \Sigma_{\pi_e} x}{x^\top \Sigma_\rho x} \right)$. Even when density ratio is infinite, this number could still be finite as it concerns subspaces on $\phi(s,a)$ rather than the whole $\mathcal{S} \times \mathcal{A}$.

To further gain its intuition, we can consider discrete MDPs and the feature mapping $\phi(s,a) \in \mathbb{R}^{|\mathcal{S}||\mathcal{A}|}$ which is a one-hot encoding vector that has zero everywhere except one at the entry corresponding to the pair $(s,a)$. In this case, the relative condition number is reduced to $\max_{(s,a)} d_P^{\pi_e}(s,a)/\rho(s,a)$.

**Theorem 6** (Error for KNRs). *Suppose $\sup_{s,a} \|\phi(s,a)\| \leq 1$, $\lambda = \Omega(1), \zeta^2 = \Omega(1), \|W^*\|_2 = \Omega(1)$ and the partial coverage $C^{\pi_e} < \infty$. With probability $1 - \delta$,*

$$\text{Err}_o \leq c_1 H^2 \left( \text{rank}^2(\Sigma_\rho) + \text{rank}(\Sigma_\rho) \log(\tfrac{c_2}{\delta}) \right) \sqrt{\frac{d_{\mathcal{S}} C^{\pi_e}}{n_o}} \cdot \log^{1/2}(1 + n_o), \qquad (2)$$

*where $c_1$ and $c_2$ are some universal constants.*

Theorem 6 suggests $\text{Err}_o$ is $\tilde{O}(H^2 \text{rank}[\Sigma_\rho]^2 \sqrt{d_{|\mathcal{S}|} C^{\pi_e}/n_o})$. In other words, when $C^{\pi_e}, \text{rank}[\Sigma_\rho]$ are small and the offline sample size $n_o$ is large enough, $\text{Err}_e$ dominates $\text{Err}_o$ in theorem 3. Again, in this case, $\text{Err}_e = O\left(H\sqrt{\ln(\mathcal{F})/n_e}\right)$, and we see that it grows linearly w.r.t horizon $H$.

Our result is distribution dependent and captures the possible low-rankness of the offline data, i.e., $\text{rank}[\Sigma_\rho]$ depends on $\rho$ and could be much smaller than the ambient dimension of feature $\phi(s, a)$. The quantity $C^{\pi_e}$ corresponds to the discrepancy measured between the batch data and expert data. This is much smaller than the worst-case concentrability coefficient: $\tilde{C} = \sup_{\pi \in \Pi} C^\pi$.

**Remark 7.** *In RL, a similar quantity has been analyzed in [31], which studies the error bound of linear FQI with pessimism. Comparing to our result only requiring partial coverage, [31, Corollary 4.5] assumes the global coverage, i.e., $\Sigma_\rho$ is full-rank, which is stronger than $\tilde{C} < \infty$.*

**GPs**   Now we specialize our main theorem to non-parametric GP models. For simplicity, following [63], we assume $\mathcal{S} \times \mathcal{A}$ is a compact space. We also suppose the following. Recall $x := (s, a)$.

**Assumption 8.** $k(x, x) \leq 1, \forall x \in \mathcal{S} \times \mathcal{A}$. $k(\cdot, \cdot)$ *is a continuous and positive semidefinite kernel.*

Under the assumption 8, we can use Mercer's theorem [73], which shows that there exists a set of pairs of eigenvalues and eigenfunctions $\{\mu_i, \psi_i\}_{i=1}^\infty$, where $\int \rho(x)\psi_i(x)\psi_i(x)dx = 1$ for all $i$ and $\int \rho(x)\psi_i(x)\psi_j(x)dx = 0$ for $i \neq j$. Eigenfunctions and eigenvalues essentially defines an infinite-dimensional feature mapping $\phi(x) := [\sqrt{\mu_1}\psi_1(x), \ldots, \sqrt{\mu_\infty}\psi_\infty(x)]^\top$. Here, $k(x, x) = \phi(x)^\top \phi(x)$, and any function $f \in \mathcal{H}_k$ can be represented as $f(\cdot) = \alpha^\top \phi(\cdot)$. Note that the eigenvalues and eigenfunctions are defined w.r.t the offline data $\rho$, thus our result here is still distribution dependent rather than a worst case analysis which often appears in online RL/IL settings [63; 32; 78; 17].

Assume eigenvalues $\{\mu_1, \ldots, \mu_\infty\}$ are in non-increasing order, we define the effective dimension,

**Definition 9** (Effective dimension). $d^* = \min\{j \in \mathbb{N} : j \geq B(j+1)n_o/\zeta^2\}$, $B(j) = \sum_{k=j}^\infty \mu_k$.

The effective dimensions $d^*$ is widely used and calculated for many kernels [85; 7; 72; 28]. In finite-dimensional linear kernels $\{x \mapsto a^\top \phi(x); a \in \mathbb{R}^d\}$ ($k(x, x) = \phi^\top(x)\phi(x)$), we have $d^* \leq \text{rank}[\Sigma_\rho]$. Thus, $d^*$ is considered to be a natural extension of $\text{rank}[\Sigma_\rho]$ to infinite-dimensional models.

**Theorem 10** (Error for GPs). *Let $\Sigma_{\pi_e} = \mathbb{E}_{x \sim d_P^{\pi_e}}[\phi(x)\phi(x)^\top], \Sigma_\rho = \mathbb{E}_{x \sim \rho}[\phi(x)\phi(x)^\top]$. Suppose assumption 8, $\zeta^2 = \Omega(1)$ and the partial coverage $C^{\pi_e} = \sup_{\|x\|_2 \leq 1}(x^\top \Sigma_{\pi_e} x / x^\top \Sigma_\rho x) < \infty$. With probability $1 - \delta$,*

$$\text{Err}_o \leq c_1 H^2 \left((d^*)^2 + d^* \log(c_2/\delta)\right) \sqrt{\frac{d_\mathcal{S} C^{\pi_e}}{n_o}} \cdot \sqrt{\log^3(c_2 d_\mathcal{S} n_o/\delta) \log(1 + n_o)}, \qquad (3)$$

*where $c_1, c_2$ are universal constants.*

The theorem suggests that $\text{Err}_o$ is $\tilde{O}(H^2 d^{*2}\sqrt{d_\mathcal{S} C^{\pi_e}/n_o})$. Thus, when $C^{\pi_e}, d^*$ are not so large and $n_o$ is large enough, $\text{Err}_e$ asymptotically dominates $\text{Err}_o$ in theorem 3 (again $\text{Err}_e$ is linear in $H$).

While we defer the detailed proof of the above theorem to Appendix C.3, we highlight some techniques we used here. The analysis is reduced to how to bound the information gain $\mathcal{I}_{n_o}$ and $\mathbb{E}_{x \sim d_P^{\pi_e}}[k_{n_o}(x, x)]$. In both cases, we analyze them into two steps: transforming them into the variational representation and then bounding them via the uniform low with localization (Lemma 39).

**Remark 11** (Implication to Offline RL). *As related literature, in model-free offline RL, [70; 21] obtain the finite-sample error bounds using nonparametric models. Though their bounds can be characterized by the effective dimension, their bounds assume full coverage, i.e., $\max_{(s,a)} 1/\rho(s, a) < \infty$.*

## 5   Practical Algorithm

In this section we instantiate a practical version of `MILO` using neural networks for the model class $\mathcal{P}$ and policy class $\Pi$. We use the Maximum Mean Discrepancy (MMD) with a Radial Basis Function kernel as our discriminator class $\mathcal{F}$. Note using MMD as our discrepancy measure allows us to compute the exact maximum discriminator $\text{argmax}_{f \in \mathcal{F}}$ in closed form (and is detailed in appendix). We use a KL-based trust-region formulation for incremental policy update inside the learned model $\hat{P}$. Based on Eq. (1), we first formalize the following constrained optimization framework:

$$\min_{\pi \in \Pi} \max_{f \in \mathcal{F}} \left(\mathbb{E}_{(s,a) \sim d_{\hat{P}}^\pi}[f(s, a) + b(s, a)] - \mathbb{E}_{(s,a) \sim \mathcal{D}_e}[f(s, a)]\right) \text{ s.t. } \mathbb{E}_{(s,a) \sim \mathcal{D}_e}[\ell(a, s, \pi)] \leq \delta,$$

where $\ell : \mathcal{A} \times \mathcal{S} \times \Pi \mapsto \mathbb{R}$ is a loss function (e.g., negative log-likelihood or any supervised learning loss one would use in BC). Essentially, since we have $\mathcal{D}_e$ available, we use it together with any

---

**Algorithm 2** A practical instantiation of `MILO`

---

1: **Require**: expert dataset $\mathcal{D}_e = \{s, a\}$, offline dataset $\mathcal{D}_o := \{s, a, s'\}$
2: Train an ensemble of neural network models $\{\hat{g}_1, \ldots, \hat{g}_n\}$ where each $P_i$ starts with different random initialization;
3: Set bonus $b(s, a) = \max_{i,j} \|g_i(s, a) - g_j(s, a)\|_2$ and initialize $\pi_{\theta_0}$.
4: **for** $t = 0 \to T - 1$ **do**
5: $\quad w_t = \arg\max_{\|w\|_2 \leq 1} w^\top \left( \mathbb{E}_{(s,a) \sim d_{\hat{P}}^\pi}[\phi(s, a)] - \mathbb{E}_{(s,a) \sim \mathcal{D}_e}[\phi(s, a)] \right), f_t(s, a) := w_t^\top \phi(s, a)$

6: $\quad \theta_{t+1} = \theta_t - \eta F_{\theta_t}^{-1} \left( \mathbb{E}_{(s,a) \sim d_{\hat{P}}^{\pi_{\theta_t}}} \left[ \nabla \ln \pi_{\theta_t}(a|s) A_{\hat{P}, f_t+b}^{\pi_{\theta_t}}(s, a) \right] + \lambda \mathbb{E}_{(s,a) \sim \mathcal{D}_e} \left[ \nabla \ell(a, s, \pi_{\theta_t}) \right] \right)$

7: **end for**

---

supervised learning loss to constrain the policy hypothesis space. Note for a deterministic expert $\pi_e$, the expert policy is always a feasible solution. Thus adding this constraint reduces the complexity of the policy class but does not eliminate the expert policy, and our analysis in Section 4 still applies.

In our practical instantiation, we replace the hard constraint instead by a Lagrange multiplier, i.e. we use the behavior cloning objective as a regularization term when solving the min-max problem:

$$\min_{\pi \in \Pi} \max_{f \in \mathcal{F}} \left[ \mathbb{E}_{(s,a) \sim d_{\hat{P}}^\pi} \left( f(s, a) + b(s, a) \right) - \mathbb{E}_{(s,a) \sim \mathcal{D}_e}[f(s, a)] \right] + \lambda \cdot \mathbb{E}_{(s,a) \sim \mathcal{D}_e}[\ell(a, s, \pi)].$$

Given policy $\pi_{\theta_t}$ ($\theta_t$ denotes the parameters), we update the discriminator $f_t$. Then, with a fixed $f_t$, in order to update policy $\pi$ we use NPG as in line 6 in Algorithm 2, where $A_{\hat{P}, f+b}^{\pi_\theta}$ is the disadvantage function of $\pi_\theta$ and $F_{\theta_t} := \mathbb{E}_{(s,a) \sim d_{\hat{P}}^{\pi_{\theta_t}}} \left[ \nabla \ln \pi_{\theta_t}(a|s) \nabla \ln \pi_{\theta_t}(a|s)^\top \right]$ is the fisher information matrix.

## 6 Experiments

We aim to answer the following questions with our experiments: (1) How does `MILO` perform relative to other offline IL methods, (2) What is the impact of pessimism on `MILO`'s performance? (3) How does the behavior policy's coverage impact `MILO`'s performance? (4) How does `MILO`'s result vary when we increase the number of samples drawn from the expert policy?

We evaluate `MILO` on five environments from OpenAI Gym [11] simulated with MuJoCo [67]: `Hopper-v2`, `Walker2d-v2`, `HalfCheetah-v2`, `Ant-v2`, and `Humanoid-v2`. We compare `MILO` against the following baselines: (1) ValueDICE [38], a state-of-the-art off-policy IL method modified for the offline IL setting; (2) BC on the expert dataset; and (3) BC on both the offline and expert dataset.

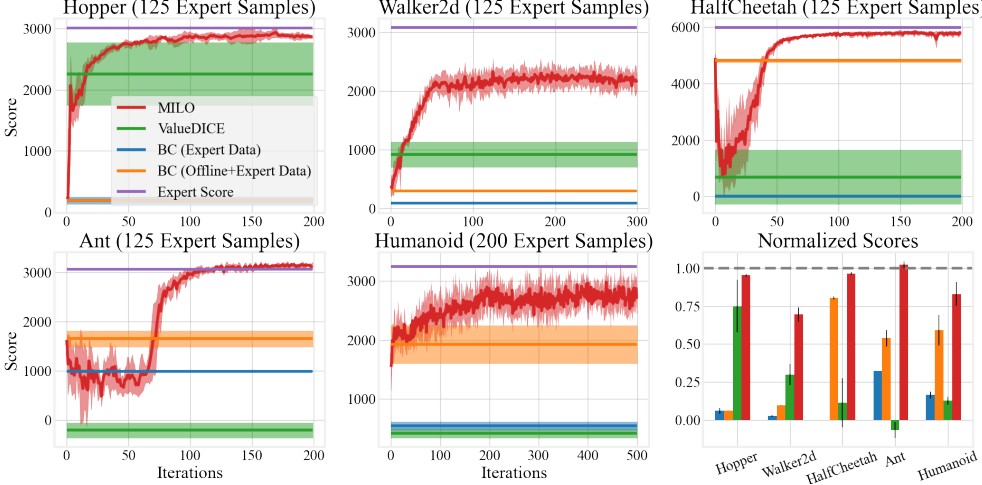

Figure 2: Learning curves across five seeds for `MILO` plotted against the best performance of BC after 1000 epochs of training on the expert/offline+expert data and the best performance of ValueDICE after 10 thousand iterations. The bottom right bar graph shows the expert performance normalized scores where we plot the performance at the last iteration for `MILO`.

For the expert dataset, we first train expert policies and then randomly sample $(s, a)$-pairs from a pool of 100 expert trajectories collected from these expert policies. We randomly sample to create very small expert $(s, a)$-pair datasets where BC struggles to learn. Note that BC is effective at imitating the expert for MuJoCo tasks even with a single trajectory; prior works [27; 37; 38] have used similar sub-sampling strategies to create expert datasets to make it harder for BC to learn. The offline datasets are collected building on prior Offline RL works [76; 34]; each dataset contains 1 million samples from the environment. We first train behavior policies with mean performances often less than half of the expert performance (Table 1, column 2). All results are averaged over five random seeds. See appendix for details on hyperparameters, environments, and dataset composition.

Table 1: Performance for expert and behavior policy used to collect expert and offline datasets respectively.

| Environment | Expert Performance | Behavior Performance |
|---|---|---|
| Hopper-v2 | 3012 | 752 (25%) |
| Walker2d-v2 | 3082 | 1383 (45%) |
| HalfCheetah-v2 | 5986 | 3972 (66%) |
| Ant-v2 | 3072 | 1208 (40%) |
| Humanoid-v2 | 3248 | 1505 (46%) |

## 6.1 Evaluation on MuJoCo Continuous Control Tasks

Figure 2 presents results comparing `MILO` against benchmarks. `MILO` is able to achieve close to expert level performance on three out of the five environments and outperforms both BC and ValueDICE on all five environments. Both `MILO` and ValueDICE were warmstarted with one epoch of BC on the offline dataset. We significantly outperform BC's performance when trained on the expert dataset, suggesting `MILO` indeed mitigates covariate shift through the use of a static offline dataset of $(s, a)$-pairs. BC on both the offline and expert dataset does improve the performance, but this still cannot successfully imitate the expert since BC has no way of differentiating random/sub-optimal trajectories from the expert samples. ValueDICE, on the other hand, does explicitly aim to imitate the expert samples; however, in theory, it would require either the offline data (i.e. the replay buffer) or the expert samples to have full coverage over the state-action space. Since our offline dataset is mainly collected from a sub-optimal behavior policy and our expert samples are from a high quality expert, neither our offline nor our expert dataset is likely to have full coverage globally; thus potentially hurting the performance of algorithms like ValueDICE. Note that `MILO` is still able to perform reasonably well across environments even with these offline and expert datasets.

## 6.2 Ablation

**Impact of Pessimism** Figure 3 (Left 2) presents `MILO`'s performance on two representative environments with and without pessimism (i.e., setting penalty to be zero) added to the imitation objective. Pessimism stabilizes and improves the final performance for `MILO`. In general, `MILO` consistently outperforms benchmarks and/or achieves expert level performance for a given set of hyperparameters. See the Appendix for evaluation on other environments.

**Behavior with more expert samples** We investigate whether `MILO` is able to achieve expert performance with more expert samples in the two environments (walker and humanoid) that it did not

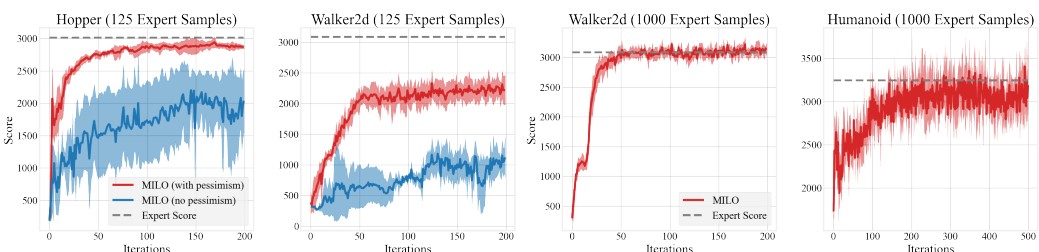

Figure 3: (Left 2) Learning curves for Hopper and Walker2d with (red) and without (blue) pessimism. `MILO` generally performs worse without pessimism. (Right 2) Learning curve for Walker2d and Humanoid with more expert samples.

solve with very small expert datasets in Figure 2. Figure 3 (Right 2) shows that with one trajectory worth of expert samples, MILO is able to achieve expert performance on walker and humanoid.

**Impact of Coverage** As our analysis suggests, MILO's performance degrades as the offline data's coverage over the expert's state-action space decreases. We use the behavior policy's value as a surrogate for lower coverage, i.e. a lower value suggests lower coverage. We generate two additional offline datasets for each environment by lowering the performance of the behavior policy. The three datasets are: (1) the original offline datasets used in Table 1 ($\approx 25\%$ for Hopper-v2 and $\approx 50\%$ for others); (2) ones

Table 2: Expert performance normalized scores on three different offline datasets collected from behavior policies with approximately 50%, 25%, and random performance relative to the expert.

| Environment | $\approx 50\%$ | $\approx 25\%$ | Random |
|---|---|---|---|
| Hopper-v2 | $0.95 \pm 0.01$ | $0.66 \pm 0.33$ | $0.42 \pm 0.36$ |
| Walker2d-v2 | $0.72 \pm 0.02$ | $0.27 \pm 0.06$ | $0.23 \pm 0.12$ |
| HalfCheetah-v2 | $0.96 \pm 0.01$ | $0.01 \pm 0.02$ | $0.01 \pm 0.02$ |
| Ant-v2 | $1.02 \pm 0.02$ | $0.99 \pm 0.01$ | $0.21 \pm 0.52$ |
| Humanoid-v2 | $0.88 \pm 0.10$ | $0.72 \pm 0.03$ | $0.08 \pm 0.01$ |

that have roughly half the performance of (1) (12% for Hopper-v2 and $\approx 25\%$ for others); and (3) ones collected from a random behavior policy (Random). Table 2 shows that MILO performs reasonably on three environments even with a lower coverage dataset (second column) and achieves more than 20% of the expert performance on three environment even with the Random dataset.

## 7 Conclusion

MILO investigates how to mitigate covariate shift in IL using an offline dataset of environment interactions that has partial coverage of the expert's state-action space. We show the effectiveness of MILO both in theory and in practice. In future works, we hope to scale to image-based control.

We want to highlight the potential negative societal/ethical impacts our work. An IL algorithm is only as good as the expert that it is imitating, not only in terms of performance but also with regards to the negative biases and intentions that the demonstrator has. When designing real-world experiments/applications for MILO we believe the users should do their due diligence on removing any negative bias or malicious intent in the demonstrations that they provide.

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
