# Contents

# A  Penalty Designs

We show that the penalty design in section 4 is valid, i.e, the model is well-calibrated for tabular MDPs, KNRs, and GPs.

## A.1  Tabular models

**Lemma 12.** *With probability* $1 - \delta$,

$$\|\hat{P}(\cdot|s,a) - P(\cdot|s,a)\|_1 \leq \sqrt{\frac{|\mathcal{S}|\log 2 + \log(2|\mathcal{S}||\mathcal{A}|/\delta)}{2\{N(s,a) + \lambda\}}} + \frac{\lambda}{N(s,a) + \lambda} \quad \forall(s,a) \in \mathcal{S} \times \mathcal{A}.$$

*Proof.* When $N(s,a) > 0$, we use the concentration inequality of discrete distributions [30]. Then, with probability $1 - \delta$,

$$\left\|\frac{N(\cdot|s,a)}{N(s,a)} - P(\cdot|s,a)\right\|_1 \leq \sqrt{\frac{|\mathcal{S}|\log 2 + \log(2|\mathcal{S}||\mathcal{A}|/\delta)}{2N(s,a)}} \quad \forall(s,a) \in \{(s,a) : N(s,a) > 0\}.$$

Thus, noting $0 < N(s,a)/(N(s,a) + \lambda) < 1$, with probability $1 - \delta$, we have $\forall(s,a) \in \{(s,a) : N(s,a) > 0\}$,

$$\left\|\frac{N(\cdot|s,a)}{(N(s,a) + \lambda)} - P(\cdot|s,a) \times \frac{N(s,a)}{N(s,a) + \lambda}\right\|_1 \leq \sqrt{\frac{|\mathcal{S}|\log 2 + \log(2|\mathcal{S}||\mathcal{A}|/\delta)}{2\{N(s,a) + \lambda\}}}. \tag{4}$$

Besides, the above inequality is still well-defined and holds including the case $N(s,a) = 0$. Thus, with probability $1 - \delta$, we have $\forall(s,a) \in \mathcal{S} \times \mathcal{A}$, we have eq. (4).

Recall the estimator $\hat{P}$ is $N(s'|s,a)/(N(s,a) + \lambda)$. Therefore,

$$\|\hat{P}(\cdot|s,a) - P(\cdot|s,a)\|_1 \leq \left\|\hat{P}(\cdot|s,a) - P(\cdot|s,a) \times \frac{N(s,a)}{N(s,a) + \lambda}\right\|_1 + \left\|P(\cdot|s,a) - P(\cdot|s,a) \times \frac{N(s,a)}{N(s,a) + \lambda}\right\|_1$$

$$\leq \sqrt{\frac{|\mathcal{S}|\log 2 + \log(2|\mathcal{S}||\mathcal{A}|/\delta)}{2\{N(s,a) + \lambda\}}} + \frac{\lambda}{N(s,a) + \lambda}.$$

This concludes the proof. $\qquad \square$

## A.2  KNRs

In KNRs, the ground truth model is $s' = W^*\phi(s,a) + \epsilon, \epsilon \sim \mathcal{N}(0, \zeta^2\mathbf{I})$, where $s \in \mathbb{R}^{d_\mathcal{S}}, a \in \mathbb{R}^{d_\mathcal{A}}, \phi : \mathcal{S} \times \mathcal{A} \to \mathbb{R}^d$. We define

$$\|\phi(s,a)\|_{\Sigma_{n_o}^{-1}} := \phi^\top(s,a)\Sigma_{n_o}^{-1}\phi(s,a).$$

**Lemma 13.** *With probability at least* $1 - \delta$, *we have:*

$$\left\|\hat{P}(\cdot|s,a) - P(\cdot|s,a)\right\|_1 \leq \min\left\{\frac{\beta_{n_o}}{\zeta}\|\phi(s,a)\|_{\Sigma_{n_o}^{-1}}, 2\right\} \quad \forall(s,a) \in \mathcal{S} \times \mathcal{A},$$

*where*

$$\beta_{n_o} = \sqrt{2\lambda\|W^\star\|_2^2 + 8\zeta^2\left(d_\mathcal{S}\ln(5) + \ln(1/\delta) + \bar{\mathcal{I}}_{n_o}\right)}, \quad \bar{\mathcal{I}}_{n_o} = \ln\left(\det(\Sigma_{n_o})/\det(\lambda\mathbf{I})\right).$$

*Proof.* The proof directly follows the confidence ball construction and proof from [32]. Specifically, from Lemma B.5 in [32], we have that with probability at least $1 - \delta$,

$$\left\|\left(\widehat{W} - W^\star\right)(\Sigma_{n_o})^{1/2}\right\|_2^2 \leq \beta_{n_o}^2.$$

Thus, with Lemma 34, we have:

$$\left\|\hat{P}(\cdot|s,a) - P(\cdot|s,a)\right\|_1 \leq \frac{1}{\zeta}\left\|(\widehat{W} - W^\star)\phi(s,a)\right\|_2$$

$$\leq \left\|(\widehat{W} - W^\star)(\Sigma_{n_o})^{1/2}\right\|_2\|\phi(s,a)\|_{\Sigma_{n_o}^{-1}}/\zeta \leq \frac{\beta_{n_o}}{\zeta}\|\phi(s,a)\|_{\Sigma_{n_o}^{-1}}.$$

This concludes the proof. $\qquad \square$

### A.3 Gaussian processes

Let $\mathcal{H}_k$ be the RKHS with the kernel $k(\cdot, \cdot)$. We denote the associated norm and inner product by $\| \cdot \|_k$ and $\langle \cdot, \cdot \rangle_k$. In GPs, the ground truth model is defined as $s' = g^*(s, a) + \epsilon, \epsilon \sim \mathcal{N}(0, \zeta^2 \mathbf{I})$, where $g^*$ belongs to an RKHS $\mathcal{H}_k$.

**Lemma 14.** *With probability* $1 - \delta$,

$$\|\hat{P}(\cdot|s,a) - P(\cdot|s,a)\|_1 \leq \min\left(\frac{\beta_{n_o}}{\zeta}\sqrt{k_{n_o}((s,a),(s,a))}, 2\right) \quad \forall(s,a) \in \mathcal{S} \times \mathcal{A},$$

*and*

$$\beta_{n_o} = \sqrt{d_{\mathcal{S}}\{2 + 150\log^3(d_{\mathcal{S}}n_o/\delta)\mathcal{I}_{n_o}\}}, \quad \mathcal{I}_{n_o} = \log(\det(\mathbf{I} + \zeta^{-2}\mathbf{K}_{n_o})).$$

*Proof.* Let $\hat{g}_i$ and $g^*$ be $i$-th component of $\hat{g}$ and $g^*$. We have

$$\|\hat{P}(\cdot|s,a) - P(\cdot|s,a)\|_1 \leq \frac{1}{\zeta}\|\hat{g}(s,a) - g^*(s,a)\|_2 \qquad (\text{Lemma 34})$$

$$= \frac{1}{\zeta}\left(\sum_{i=1}^{d_{\mathcal{S}}}\{\hat{g}_i(s,a) - g_i^*(s,a)\}^2\right)^{1/2}$$

$$\leq \frac{1}{\zeta}\left(\sum_{i=1}^{d_{\mathcal{S}}}k_{n_o}((s,a),(s,a))\|\hat{g}_i - g_i^*\|_{k_{n_o}}^2\right)^{1/2}.$$

$$(\text{CS inequality and } g = \langle g(\cdot), k((s,a),\cdot)\rangle_{k_{n_o}})$$

By [63, Theorem 6], with probability $1 - \delta$, we have

$$\|\hat{g}_i(s,a) - g_i^*\|_{k_{n_o}} \leq \beta_{n_o} \quad \forall i \in [1, \cdots, d_{\mathcal{S}}].$$

This concludes the statement.

$\square$

## B Proof of theorem 3

In this section, we prove Theorem 3. We also prove the RL version of Theorem 3 when the cost $c$ is given and the goal is policy optimization. Before that, we prepare several lemmas.

**Lemma 15.** *With probability* $1 - \delta$, *we have* $\forall f \in \mathcal{F}$,

$$|\mathbb{E}_{(s,a)\sim d^{\pi_e}}[f(s,a)] - \mathbb{E}_{D_e}[f(s,a)]| \leq \epsilon_{\text{stat}}, \quad \epsilon_{\text{stat}} = \sqrt{\log(2|\mathcal{F}|/\delta)/2n_e}.$$

*Proof.* From Hoeffding's inequality and a union bound over $\mathcal{F}$. $\square$

**Lemma 16** (Pessimistic Policy Evaluation 1 )**.** *Suppose Assumption 2 holds and* $\max_{f \in \mathcal{F}}\|f\|_\infty \leq 1$. *With probability at least* $1 - \delta$, $\forall \pi \in \Pi, \forall f \in \mathcal{F}$,

$$0 \leq V_{\hat{P}, f+b}^\pi - V_{P,f}^\pi.$$

*Proof of Lemma 16.* We denote the expected total cost of $\pi$ under $\hat{P}$ and cost function $f$ by $V_{\hat{P}, f:h}^\pi(s, a)$. In this proof, we condition on the event

$$\|\hat{P}(\cdot|s,a) - P(\cdot|s,a)\|_1 \leq \min(\sigma(s,a), 2) \quad \forall(s,a) \in \mathcal{S} \times \mathcal{A}.$$

We use the inductive hypothesis argument. We start from $h = H + 1$, where $V_{\hat{P}, f+b:H+1}^\pi = V_{P, f:H+1}^\pi = 0$. Assume the inductive hypothesis holds at $h + 1$, i.e,

$$0 \leq V_{\hat{P}, f+b:h+1}^\pi(s) - V_{P, f:h+1}^\pi(s), \quad \forall s \in \mathcal{S}, \forall \pi \in \Pi, \forall f \in \mathcal{F}.$$

Then, $\forall \pi \in \Pi$, $\forall f \in \mathcal{F}$,

$$Q_{P,f:h}^{\pi}(s,a) - Q_{\hat{P},f+b:h}^{\pi}(s,a)$$
$$= -b(s,a) + \mathbb{E}_{s' \sim \hat{P}(\cdot|s,a)}[V_{P,f:h+1}^{\pi}(s')] - \mathbb{E}_{s' \sim P(\cdot|s,a)}[V_{\hat{P},f+b:h+1}^{\pi}(s')]$$
$$\leq -b(s,a) + \mathbb{E}_{s' \sim \hat{P}(\cdot|s,a)}[V_{P,f:h+1}^{\pi}(s')] - \mathbb{E}_{s' \sim P(\cdot|s,a)}[V_{P,f:h+1}^{\pi}(s')]$$
$$\text{(Inductive hypothesis assumption)}$$
$$\leq -b(s,a) + H\|\hat{P}(\cdot|s,a) - P(\cdot|s,a)\|_1 \qquad\qquad (\|\mathcal{F}\|_\infty \leq 1)$$
$$\leq -H\min(\sigma(s,a),2) + H\min(\sigma(s,a),2) = 0. \qquad \text{(Bonus construction)}$$

Then, noting $Q_{P,f:h}^{\pi}(s,\pi(s)) - Q_{\hat{P},f+b:h}^{\pi}(s,\pi(s)) = V_{P,f:h}^{\pi}(s) - V_{\hat{P},f+b:h}^{\pi}(s)$, we have

$$V_{P,f:h}^{\pi}(s) - V_{\hat{P},f+b:h}^{\pi}(s) \leq 0 \quad \forall \pi \in \Pi, \forall f \in \mathcal{F}.$$

This concludes the induction step.

Then, we have

$$V_{P,f}^{\pi} - V_{\hat{P},f+b}^{\pi} = V_{P,f:1}^{\pi} - V_{\hat{P},f+b:1}^{\pi} \leq 0 \quad \forall \pi \in \Pi, \forall f \in \mathcal{F}.$$

$$\qquad\qquad\qquad\qquad\qquad\qquad\qquad\qquad\qquad\qquad\qquad\qquad\qquad\qquad\qquad \square$$

**Lemma 17** (Pessimistic Policy Evaluation 2 ). *Suppose Assumption 2 holds and $\|\mathcal{F}\|_\infty \leq 1$. With probability at least $1 - \delta$, $\forall \pi \in \Pi$, $\forall f \in \mathcal{F}$,*

$$V_{\hat{P},f+b}^{\pi} - V_{P,f}^{\pi} \leq \text{Error}, \quad \text{Error} := (3H^2 + H)\mathbb{E}_{(s,a) \sim d_P^{\pi}}[\min(\sigma(s,a),2)].$$

*Proof of Lemma 17.* In this proof, we condition on the event

$$\|\hat{P}(\cdot|s,a) - P(\cdot|s,a)\|_1 \leq \min(\sigma(s,a),2) \quad \forall(s,a) \in \mathcal{S} \times \mathcal{A}.$$

We invoke simulation Lemma 33. Then, we have $\forall \pi \in \Pi, \forall f \in \mathcal{F}$

$$V_{\hat{P},f+b}^{\pi} - V_{P,f}^{\pi} = \sum_{h=1}^{H} \mathbb{E}_{(s,a) \sim d_P^{\pi}}[b(s,a) + \mathbb{E}_{s' \sim \hat{P}(\cdot|s,a)}[V_{\hat{P},f+b:h}^{\pi}(s')] - \mathbb{E}_{s' \sim P(\cdot|s,a)}[V_{\hat{P},f+b:h}^{\pi}(s')]]$$

$$\leq \sum_{h=1}^{H} \mathbb{E}_{(s,a) \sim d_P^{\pi}}[b(s,a) + \|V_{\hat{P},f+b:h}^{\pi}\|_\infty \|\hat{P}(\cdot|s,a) - P(\cdot|s,a)\|_1]$$

$$\leq H\mathbb{E}_{(s,a) \sim d_P^{\pi}}[H\min(\sigma(s,a),2) + H(2H+1)\min(\sigma(s,a),2)]$$
$$( \|V_{\hat{P},f+b:h}^{\pi}\|_\infty \leq H(2H+1))$$

$$= (3H^2 + H)\mathbb{E}_{(s,a) \sim d_P^{\pi}}[\min(\sigma(s,a),2)].$$

Here, we use $\|V_{\hat{P},f+b:h}^{\pi}\|_\infty \leq H(2H+1)$ which is derived by $0 \leq f + b \leq 2H + 1$. $\qquad \square$

By using the above lemmas, we prove our main result.

*Proof of Theorem 3.* In this proof, we condition on the event

$$\|\hat{P}(\cdot|s,a) - P(\cdot|s,a)\|_1 \leq \min(\sigma(s,a),2),$$

which holds with probability $1 - \delta$, and the event in Lemma 15, which holds with probability $1 - \delta$.

Then, with probability $1 - 2\delta$, we have

$$
\begin{aligned}
V_{P,c}^{\hat{\pi}_{\mathrm{IL}}} - V_{P,c}^{\pi_e} &\leq V_{\hat{P},c+b}^{\hat{\pi}_{\mathrm{IL}}} - V_{P,c}^{\pi_e} && \text{(Lemma 16)} \\
&\leq H \max_{f \in \mathcal{F}} \{ \mathbb{E}_{(s,a) \sim d_{\hat{P}}^{\hat{\pi}_{\mathrm{IL}}}}[f(s,a) + b(s,a)] - \mathbb{E}_{(s,a) \sim d_P^{\pi_e}}[f(s,a)] \} && (c \in \mathcal{F}) \\
&\leq H \max_{f \in \mathcal{F}} \{ \mathbb{E}_{(s,a) \sim d_{\hat{P}}^{\hat{\pi}_{\mathrm{IL}}}}[f(s,a) + b(s,a)] - \mathbb{E}_{\mathcal{D}_e}[f(s,a)] \} + H\epsilon_{\mathrm{stats}} && \text{( Lemma 15)} \\
&\leq H \max_{f \in \mathcal{F}} \{ \mathbb{E}_{(s,a) \sim d_{\hat{P}}^{\pi_e}}[f(s,a) + b(s,a)] - \mathbb{E}_{\mathcal{D}_e}[f(s,a)] \} + H\epsilon_{\mathrm{stats}} \\
& && (\pi_e \in \Pi \text{ and the definition of } \hat{\pi}_{\mathrm{IL}}) \\
&\leq H \max_{f \in \mathcal{F}} \{ \mathbb{E}_{(s,a) \sim d_{\hat{P}}^{\pi_e}}[f(s,a) + b(s,a)] - \mathbb{E}_{(s,a) \sim d_P^{\pi_e}}[f(s,a)] \} + 2H\epsilon_{\mathrm{stats}} \\
& && \text{( Lemma 15)} \\
&\leq \max_{f \in \mathcal{F}} \{ V_{\hat{P},f+b}^{\pi_e} - V_{P,f}^{\pi_e} \} + 2H\epsilon_{\mathrm{stats}} \\
&\leq (3H^2 + H) \mathbb{E}_{(s,a) \sim d_P^{\pi_e}}[\min(\sigma(s,a), 2)] + 2H\epsilon_{\mathrm{stats}} && \text{(Lemma 17)} \\
&\leq (6H^2 + 2H) \mathbb{E}_{(s,a) \sim d_P^{\pi_e}}[\min(\sigma(s,a), 1)] + 2H\epsilon_{\mathrm{stats}}.
\end{aligned}
$$

This concludes the proof. $\qquad\square$

Finally, we prove the finite-sample error bounds for the RL case. Similar results are obtained in [34; 82]. We use this theorem in the next section.

**Theorem 18** (Bounds for RL). *Suppose $\pi^* \in \Pi, P \in \mathcal{P}$ and Assumption 2. With probability $1 - 2\delta$, we have*

$$
V_{P,c}^{\hat{\pi}_{\mathrm{RL}}} - V_{P,c}^{\pi^*} \leq (6H^2 + 2H) \mathbb{E}_{(s,a) \sim d_P^{\pi^*}}[\min(\sigma(s,a), 1)]. \tag{5}
$$

*Proof of Theorem 18 .*

$$
\begin{aligned}
V_{P,c}^{\hat{\pi}_{\mathrm{RL}}} - V_{P,c}^{\pi^*} &\leq V_{\hat{P},c+b}^{\hat{\pi}_{\mathrm{RL}}} - V_{P,c}^{\pi^*} && \text{(Lemma 16)} \\
&= V_{\hat{P},c+b}^{\pi^*} - V_{P,c}^{\pi^*} && (\pi^* \in \Pi \text{ and the definition of } \hat{\pi}_{\mathrm{RL}}) \\
&= (3H^2 + H) \mathbb{E}_{(s,a) \sim d_P^{\pi^*}}[\min(\sigma(s,a), 2)] && \text{(Lemma 17)} \\
&\leq (6H^2 + 2H) \mathbb{E}_{(s,a) \sim d_P^{\pi^*}}[\min(\sigma(s,a), 1)].
\end{aligned}
$$

This concludes the proof. $\qquad\square$

## C  Finite sample error bound for each model

In this section, we analyze the bound for the following models: (1) discrete MDPs, (2) KNRs, (3) GPs. All of the proofs are deferred to Section C.4. We will also discuss the implication to the RL case using theorem 18.

### C.1  Discrete MDPs

Recall $\pi_e$-concentratabiliy coefficient is defined by

$$
C^{\pi_e} = \max_{(s,a)} \frac{d_P^{\pi_e}(s,a)}{\rho(s,a)}.
$$

Then, the error is calculated as follows.

**Theorem 19** (Error of `MILO` for discrete MDPs).

- *With probability $1 - \delta$, when $\lambda = \Omega(1)$,*

$$
V_{P,c}^{\hat{\pi}_{\mathrm{IL}}} - V_{P,c}^{\pi_e} \leq \mathrm{Err}_o + \mathrm{Err}_e,
$$

$$
\mathrm{Err}_o = c_1 H^2 \log(|\mathcal{S}||\mathcal{A}|c_2/\delta) \left( \sqrt{\frac{C^{\pi_e}|\mathcal{S}|^2|\mathcal{A}|}{n_o}} + \frac{C^{\pi_e}|\mathcal{S}||\mathcal{A}|}{n_o} \right), \mathrm{Err}_e = 2H\sqrt{\frac{\log(2|\mathcal{F}|/\delta)}{2n_e}},
$$

*where $c_1$ and $c_2$ are some universal constants.*

- *With probability $1 - \delta$, when $\lambda = \Omega(1)$,*

$$V_{P,c}^{\hat{\pi}_{\mathrm{RL}}} - V_{P,c}^{\pi_e} \leq c_1 H^2 \log(|\mathcal{S}||\mathcal{A}|c_2/\delta) \left( \sqrt{\frac{C^{\pi^*}|\mathcal{S}|^2|\mathcal{A}|}{n_o}} + \frac{C^{\pi^*}|\mathcal{S}||\mathcal{A}|}{n_o} \right), \quad C^{\pi^*} = \max_{(s,a)} \frac{d_P^{\pi^*}(s,a)}{\rho(s,a)}.$$

(6)

  *where $c_1$ and $c_2$ are some universal constants.*

The quantity $C^{\pi_e}$ measures the difference of distributions between the expert and the batch data. This is much smaller than the common concentratabiliy coefficients in offline RL:

$$\max_{\pi \in \Pi} \max_{(s,a) \in \mathcal{S} \times \mathcal{A}} \frac{d_P^{\pi}(s,a)}{\rho(s,a)}, \quad \frac{1}{\min_{(s,a)} \rho(s,a)},$$

which measure the worst discrepancy between all policies in $\Pi$ and the batch data [80]. These assumptions imply $\rho$ has global coverage. We achieve this better bound via pessimism. In the RL case, the similar bound as (6) has been obtained in offline policy optimization based on FQI [54]. However, their work is limited to a tabular case. Hereafter, we will show our result is extended to more general continuous MDPs.

## C.2 KNRs

As in Proposition 13, $\sigma(s,a)$ is given by $\beta_{n_o}/\zeta\|\phi(s,a)\|_{\Sigma_{n_o}^{-1}}$. Thus, from Theorem 3, the final error bound of $\hat{V}_{P,c}^{\hat{\pi}_{\mathrm{IL}}} - V_{P,c}^{\pi_e}$ is

$$(6H^2 + 2H)\min(\mathbb{E}_{(s,a)\sim d_P^{\pi_e}}[\beta_{n_o}/\zeta\|\phi(s,a)\|_{\Sigma_{n_o}^{-1}}], 1) + 2H\sqrt{\log(2|\mathcal{F}|/\delta)/(2n_e)}.$$

Hereafter, we analyze $\beta_{n_o}$ and $\mathbb{E}_{(s,a)\sim d_P^{\pi_e}}[\|\phi(s,a)\|_{\Sigma_{n_o}^{-1}}]$.

**Analysis of information gain** First, we analyze $\beta_{n_o}$. We need to upper-bound the information gain $\bar{\mathcal{I}}_{n_o}$ in $\beta_{n_o}$. Recall $\Sigma_\rho = \mathbb{E}_{(s,a)\sim\rho}[\phi(s,a)\phi^\top(s,a)]$ and $\phi(s,a) \in \mathbb{R}^d$.

**Theorem 20** (Finite sample analysis of information gain in finite-dimensional linear models)**.** *Assume $\|\phi(s,a)\|_2 \leq 1 \, \forall(s,a) \in \mathcal{S} \times \mathcal{A}$. Let $c_1, c_2$ be universal constants.*

1. *When $\lambda = \Omega(1)$, with probability $1 - \delta$, we have*

$$\bar{\mathcal{I}}_{n_o} = \log(\det(\Sigma_{n_o}/\lambda\mathbf{I})) \leq c_1 \mathrm{rank}(\Sigma_\rho)\{\mathrm{rank}(\Sigma_\rho) + \log(c_2/\delta)\}\log(1 + n_o).$$

2. *When $\lambda = \Omega(1)$ and $\zeta^2 = \Omega(1)$, With probability $1 - \delta$, we have*

$$\beta_{n_o} \leq c_1\sqrt{\|W^*\|_2 + d_{\mathcal{S}}\mathrm{rank}(\Sigma_\rho)\{\mathrm{rank}(\Sigma_\rho) + \log(c_2/\delta)\}\log(1 + n_o)}.$$

Theorem 20 states $\bar{\mathcal{I}}_{n_o} = O(\mathrm{rank}[\Sigma_\rho]^2 \log(n_o))$. We highlight the novelty of our analysis comparing to the other literature. [60] analyze the expectation of the information gain in a fixed or random design setting. Following their discussion, we can prove

$$\mathbb{E}[\bar{\mathcal{I}}_{n_o}] \leq \mathrm{rank}(\Sigma_\rho)\log(1 + n_o)$$

as Theorem 42 by Jensen's inequality. Going beyond the expectation, we derive the finite-sample result by leveraging the variational representation and the uniform law with localization in Lemma 39. The finite-sample analysis is much harder than calculating the bound of the expectation.

The worse case of $\bar{\mathcal{I}}_{n_o}$ referred to as the maximum information gain has been often used in online learning [63; 1; 32]. From their discussion, we always have $\bar{\mathcal{I}}_{n_o} = O(d\log(n))$. Here, we show that the information gain can be upper-bounded more tightly when $\mathrm{rank}[\Sigma_\rho]^2 \leq d$ in offline RL (a random design setting). Comparing to the analysis of maximum information gain, our analysis takes the low-rankness of the design matrix $\Sigma_\rho$ into consideration by fully utilizing the random design setting assumption.

**Analysis of $\mathbb{E}_{(s,a)\sim d_P^{\pi_e}}[\|\phi(s,a)\|_{\Sigma_{n_o}^{-1}}]$ and the final bound**

Next, we analyze $\mathbb{E}_{(s,a)\sim d_P^{\pi_e}}[\|\phi(s,a)\|_{\Sigma_{n_o}^{-1}}]$.

**Theorem 21.** *Suppose* $\lambda = \Omega(1), \zeta^2 = \Omega(1), \|W^*\|_2 = \Omega(1)$. *Let* $c_1, c_2$ *be some universal constants.*

1. *With probability* $1 - \delta$,

$$\mathbb{E}_{(s,a)\sim d_P^{\pi_e}}[\|\phi(s,a)\|_{\Sigma_{n_o}^{-1}}] \le c_1 \sqrt{\frac{C^{\pi_e}\mathrm{rank}[\Sigma_\rho]\{\mathrm{rank}[\Sigma_\rho] + \log(c_2/\delta)\}}{n_o}}, \quad C^{\pi_e} = \sup_{x \in \mathbb{R}^d}\left(\frac{x^\top \Sigma_{\pi_e} x}{x^\top \Sigma_\rho x}\right),$$

   *where* $\Sigma_{\pi_e} = \mathbb{E}_{(s,a)\sim d_P^{\pi_e}}[\phi(s,a)\phi(s,a)^\top]$.

2. *With probability* $1 - \delta$,

$$V_{P,c}^{\hat{\pi}_{\mathrm{IL}}} - V_{P,c}^{\pi_e} \le \mathrm{Err}_o + \mathrm{Err}_e, \bar{R} = \mathrm{rank}[\Sigma_\rho]\{\mathrm{rank}[\Sigma_\rho] + \log(c_2/\delta)\}, \quad (7)$$

$$\mathrm{Err}_o = c_1 H^2 \min(d^{1/2}, \bar{R})\sqrt{\bar{R}}\sqrt{\frac{d_\mathcal{S} C^{\pi_e}\log(1 + n_o)}{n_o}},$$

$$\mathrm{Err}_e = 2H\sqrt{\log(2|\mathcal{F}|/\delta)/(2n_e)}.$$

3. *With probability* $1 - \delta$, *let* $C^{\pi^*} = \sup_{x \in \mathbb{R}^d}\left(\frac{x^\top \Sigma_{\pi^*} x}{x^\top \Sigma_\rho x}\right)$, $\Sigma_{\pi^*} = \mathbb{E}_{(s,a)\sim d_P^*}[\phi(s,a)\phi^\top(s,a)]$. *Then, we have*

$$V_{P,c}^{\hat{\pi}_{\mathrm{RL}}} - V_{P,c}^{\pi^*} \le c_1 H^2 \{\mathrm{rank}(\Sigma_\rho) + \log(c_2/\delta)\}\mathrm{rank}(\Sigma_\rho)\sqrt{\frac{d_\mathcal{S} C^{\pi^*}\log(1 + n_o)}{n_o}}.$$

The final bound (7) suggests $\mathrm{Err}_o$ is $\tilde{O}(H^2\mathrm{rank}[\Sigma_\rho]^2\sqrt{d_s C^{\pi_e}/n_o})$. We can also get $\tilde{O}(H^2\mathrm{rank}[\Sigma_\rho]d^{1/2}\sqrt{d_s C^{\pi_e}/n_o})$, which implies $\tilde{O}(H^2 d^{3/2}\sqrt{d_s C^{\pi_e}/n_o})$. In other words, when $C^{\pi_e}, \mathrm{rank}[\Sigma_\rho]$ are not so large and the offline sample size $n_o$ is large enough, $O(H\sqrt{\log(|\mathcal{F}|)/n_e})$ is a dominating term and the covariate shift problem in BC can be avoided since the horizon dependence is just $H$. Recall the known BC error bound is $O(H^2\sqrt{\log|\Pi|/n_e})$ [4, Chapter 14].

We see the implication of $\mathrm{Err}_o$ in more details, which also corresponds to the error of RL case. The rate regarding $n_o$ is $n_o^{-1/2}$, which is the standard rate in parametric regression. Besides, we can see the bound depends on $\mathrm{rank}[\Sigma_\rho], C^{\pi_e}$. Importantly, since we always have $\mathrm{rank}[\Sigma_\rho] \le d$, our final bound captures the possible low-rankness of the batch data. The quantity $C^{\pi_e}$ corresponds to $\pi_e$-concentrability coefficient (*-concentrability in the RL case). This is much smaller than the worst case concentrability coefficients:

$$\sup_{\pi \in \Pi} C^\pi, \quad \tilde{C} = \sup_{(s,a)} \|\phi(s,a)\|_2^2 \|\Sigma_\rho^{-1}\|_2.$$

Finally, we note the technical novelty by comparing it to the techniques developed in the offline RL literature. A quantity that is similar to $\mathbb{E}_{(s,a)\sim d_P^{\pi_e}}[\|\phi(s,a)\|_{\Sigma_{n_o}^{-1}}]$ has been analyzed in [31][6], which studies the error bound of FQI with pessimism in linear MDPs. [31, Corollary 4.5] assumes that full coverage, i.e., $\Sigma_\rho$ is full-rank and has lower bounded eigenvalues. Also the number of offline samples $n_0$ depends on the smallest eigenvalue. Our analysis just uses partial coverage with the refined concept of relative condition number and thus does not require the full rank assumption on $\Sigma_\rho$. Moreover, our bound is distribution dependent, i.e., it depends on $\mathrm{rank}[\Sigma_\rho]$ rather than the ambient dimension of the feature vector $\phi$. Thus the bound is much tighter for benign cases where the offline data from $\rho$ happens to concentrate on a low-dimensional subspace. Beyond model-based offline RL literature, one can potentially adapt the model-free offline policy evaluation results (e.g., [20; 74]) with linear function approximation to offline policy optimization (without pessimism). Such model-free results will also incur $\sup_{\pi \in \Pi} C^\pi, \tilde{C}$ and the ambient dimension $d$, instead of much more refined quantities $C^{\pi_e}$ and $\mathrm{rank}[\Sigma_\rho]$.

---

[6]They analyze $\mathbb{E}_{(s,a)\sim d_P^{\pi^*}}[\|\phi(s,a)\|_{\Sigma_{n_o}^{-1}}]$, which also appears in our RL result theorem 21.

## C.3 Gaussian processes

In this section, we give details on GPs. Note that prior works on model-free and model-based offline IL do not have results for infinite-dimensional non-parametric models. Thus our techniques developed in this section are new and relevant even to the offline RL literature—a point that we will return to at the end of this section.

From Theorem 3, the final error is

$$(6H^2 + 2H)\min(\mathbb{E}_{x \sim d_P^{\pi_e}}[\beta_{n_o}/\zeta\sqrt{k_{n_o}(x,x)}], 1) + 2H\sqrt{\log(2|\mathcal{F}|/\delta)/(2n_e)}.$$

where $x = (s, a)$. Hereafter, we analyze $\beta_{n_o}$ and $\mathbb{E}_{x \sim d_P^{\pi_e}}[\sqrt{k_{n_o}(x,x)}]$. Before going into the details, we repeat several important notations below.

In this section, following [63], for simplicity, we suppose the following:

**Assumption 22.** $k(x,x) \leq 1, \forall x \in \mathcal{S} \times \mathcal{A}$. $k(\cdot, \cdot)$ *is a continuous and positive semidefinite kernel.* $\mathcal{S} \times \mathcal{A}$ *is a compact space.*

Recall we denote $x := (s, a)$ and we have orthonormal eigenfunctions and eigenvalues $\{\psi_i, \mu_i\}_{i=1}^{\infty}$ by Mercer's theorem. We denote the feature mapping $\phi(x) := [\sqrt{\mu}_1\psi_1(x), \dots, \sqrt{\mu_\infty}\psi_\infty(x)]^\top$.

Assume eigenvalues $\{\mu_1, \dots, \mu_\infty\}$ is in non-increasing order, we recall the effective dimension:

$$d^* = \min\{j \in \mathbb{N} : j \geq B(j+1)n_o/\zeta^2\}, \ B(j) = \sum_{k=j}^{\infty}\mu_k.$$

We also introduce the empirical version of $d^*$, where $\hat{\mu}_i$ are eigenvalues of the gram matrix $\mathbf{K}_{n_o}$.

**Definition 23** (Empirical effective dimension). $\hat{d} = \min\{j \in \mathbb{N} : j \geq B(j+1)/\zeta^2, \hat{B}(j) = \sum_{k=j}^{n_o}\hat{\mu}_k\}$.

Hereafter, for simplicity, we treat $\zeta^2 = 1$, that is, $\zeta^2 = \Omega(1)$. Then, since $n_o \leq B(n_o+1)n_o/\zeta^2$, we have $d^* \leq n_o$.

The effective dimensions $\hat{d}$ and $d^*$ are widely used in machine learning literature. The first quantity $d^*$ is often referred to as the degree of freedom [85; 7]. In finite-dimensional linear kernels $\{x \mapsto a^\top\phi(x), a \in \mathbb{R}^d\}$ $(k(x,x) = \phi^\top(x)\phi(x))$, $d^*$ is $\mathrm{rank}[\mathbb{E}_{x \sim \rho}[\phi(x)\phi^\top(x)]]$. Thus, $d^*$ is considered to be a natural extension of $\mathrm{rank}[\mathbb{E}_{x \sim \rho}[\phi(x)\phi^\top(x)]]$ to infinite-dimensional models. The worst case of the second quantity:

$$\max_{\{x_1 \in \mathcal{S} \times \mathcal{A}, \cdots, x_{n_o} \in \mathcal{S} \times \mathcal{A}\}} \hat{d}$$

is often used in online learning literature [72; 28]. Up to logarithmic factors, it is equal to the maximum information gain [63]:

$$\max_{\{x_1 \in \mathcal{S} \times \mathcal{A}, \cdots, x_{n_o} \in \mathcal{S} \times \mathcal{A}\}} \log\det(\mathbf{I} + \mathbf{K}_{n_o}).$$

as shown in [13; 72]. Importantly, as we will see soon since our setting is offline (a random design setting), $\hat{d}$ can be upper-bounded much tightly than their analysis.

**Analysis of information gain** With the above in mind, we first analyze $\beta_{n_o}$. To do that, we need to bound the information gain $\mathcal{I}_{n_o}$. From [60, Leemma 1], we can easily prove

$$\mathbb{E}[\mathcal{I}_{n_o}] \leq \log(1 + n_o)d^*.$$

as in Theorem 43. Going beyond the expectation, we derive the finite-sample error bound.

**Theorem 24** (Finite sample analysis of information gain in infinite-dimensional models). *Suppose Assumption 22. Let $c_1$ and $c_2$ be universal constants.*

*1. We have*

$$\mathcal{I}_{n_o} = \log(\det(\mathbf{I} + \zeta^{-2}\mathbf{K}_{n_o})) \leq 2\hat{d}\{\log(1 + n_o/\zeta^2) + 1\}. \tag{8}$$

2. *When $\zeta^2 = \Omega(1)$, with probability $1 - \delta$,*

$$\mathcal{I}_{n_o} = \log(\det(\mathbf{I} + \zeta^{-2}\mathbf{K}_{n_o})) \leq c_1\{d^* + \log(c_2/\delta)\}d^* \log(1 + n_o).$$

3. *When $\zeta^2 = \Omega(1)$, with probability $1 - \delta$,*

$$\beta_{n_o} \leq c_1\sqrt{d_\mathcal{S}\log^3(c_2 d_\mathcal{S} n_o/\delta)\{d^* + \log(c_2/\delta)\}d^* \log(1 + n_o)}.$$

Theorem 24 states $\mathcal{I}_{n_o} = O((d^*)^2\log(n_o))$. Our bound in the offline (a random design) setting can be much tighter compared to the online setting, that is, the known upper bound of maximum information gain in [63] though we can always use this as the bound of $\mathcal{I}_{n_o}$ with probability 1. We can see this situation in linear kernels as we see in the previous section. In $d$-linear dimensional linear kernels, the maximum information gain is $d$. On the other hand, $\{d^*\}^2 = \mathrm{rank}[\Sigma_\rho]^2$ can be much smaller than $d$.

**Analysis of learning curves and the final bound**   We bound $\mathbb{E}_{x \sim d_P^{\pi_e}}[\sqrt{k_{n_o}(x, x)}]$, where

$$k_{n_o}(x, x') = k(x, x') - \bar{k}_{n_o}(x)^\top(\mathbf{K}_{n_o} + \zeta^2\mathbf{I})^{-1}\bar{k}_{n_o}(x'), \{x_i\}_{i=1}^{n_o} \sim \rho(x).$$

where $x = (s, a)$.

Recall the definition of eigenvalues $\{\mu_i\}$ and eigenfunctions $\{\psi_i\}$ (which are orthonormal), we define the feature mapping $\phi(x) = [\sqrt{\mu_1}\psi_1(x), \ldots, \sqrt{\mu_\infty}\psi_\infty(x)]^\top$. Denote $\Phi \in \mathbb{R}^{n_o \times \infty}$ as a matrix where each row of $\Phi$ corresponds to $\phi(x_i)$. Since $k(x, x') = \phi(x)^\top\phi(x')$, we can rewrite the kernel $k_{n_o}(x, x)$ as follows:

$$\begin{aligned} k_{n_o}(x, x) &= \phi(x)^\top\phi(x) - \phi(x)^\top\Phi^\top\left(\Phi\Phi^\top + \zeta^2\mathbf{I}\right)^{-1}\Phi\phi(x) \\ &= \phi(x)^\top\left[\mathbf{I} - \Phi^\top\left(\Phi\Phi^\top + \zeta^2\mathbf{I}\right)^{-1}\Phi\right]\phi(x) \\ &= \phi(x)^\top\left(\mathbf{I} + \zeta^{-2}\Phi^\top\Phi\right)^{-1}\phi(x) \\ &= \phi(x)^\top\Sigma_{n_o}^{-1}\phi(x), \end{aligned}$$

where $\Sigma_{n_o} := \mathbf{I} + \zeta^{-2}\sum_{i=1}^{n_o}\phi(x_i)\phi(x_i)^\top$, and we use matrix inverse lemma in the third equality. Note the infinite-dimensional inverse lemma is formalized in the proof.

Now we can use the relative condition number definition and Lemma 30 for a distribution change, i.e.,

$$\mathbb{E}_{x \sim d_P^{\pi_e}}[\sqrt{k_{n_o}(x, x)}] \leq \sqrt{\mathbb{E}_{x \sim d_P^{\pi_e}}[k_{n_o}(x, x)]}$$

$$= \sqrt{\mathrm{tr}\left(\mathbb{E}_{x \sim d_P^{\pi_e}}\phi(x)\phi(x)^\top\Sigma_{n_o}\right)} \leq \sqrt{C^{\pi_e}\mathrm{tr}\left(\mathbb{E}_{x \sim \rho}\phi(x)\phi(x)^\top\Sigma_{n_o}\right)} = \sqrt{C^{\pi_e}\mathbb{E}_{x \sim \rho}k_{n_o}(x, x)},$$

where

$$C^{\pi_e} = \sup_{\|x\|_2 \leq 1}\frac{x\Sigma_{\pi_e}x}{x\Sigma_\rho x}, \quad \Sigma_{\pi_e} = \mathbb{E}_{x \sim d_P^{\pi_e}}[\phi(x)\phi(x)^\top], \quad \Sigma_\rho = \mathbb{E}_{x \sim \rho}[\phi(x)\phi(x)^\top].$$

Now we only need to focus on analyzing $\mathbb{E}_{x \sim \rho}[k_{n_o}(x, x)]$.

Before proceeding to the analysis, we introduce the critical radius [9]. Given some function class $\mathcal{F}$, consider the localized population Rademacher complexity:

$$\mathcal{R}_n(\delta; \mathcal{F}) = \mathbb{E}\left[\sup_{f \in \mathcal{F}, \mathbb{E}_{x \sim \rho}[f^2(x)] \leq \delta}\left|\frac{1}{n_o}\sum_{i=1}^{n_o}\epsilon_i f(x_i)\right|\right]$$

where $\{x_i\}$ are i.i.d samples following $\rho(x)$ and $\{\epsilon_i\}$ are i.i.d Rademacher variables taking values in $\{-1, +1\}$ equiprobably, independent of the sequence $\{x_i\}$. The critical radius is defined as the minimum solution to

$$\mathcal{R}_n(\xi; \mathcal{F}) \leq \xi^2/b$$

w.r.t $\xi$ where $b$ is a value s.t. $\|\mathcal{F}\|_\infty \leq b$.

**Theorem 25.** *Suppose [Assumption 22]. Let $c_1$ and $c_2$ be universal constants.*

1. *Let $\delta_{n_o}$ be the critical radius of the function class $\{f : f \in \mathcal{H}_k, \|f\|_k \leq 1\}$. With probability $1 - \delta$,*

$$\mathbb{E}_{x \sim d_P^{\pi_e}}[\sqrt{k_{n_o}(x,x)}] \leq c_1 \zeta \delta'_{n_o} \sqrt{C^{\pi_e} d^*},$$

*where $\delta'_{n_o} = \delta_{n_o} + \sqrt{\log(c_2/\delta)/n_o}$.*

2. *Assume $\zeta^2 = \Omega(1)$. With probability $1 - \delta$,*

$$\delta_{n_o} \leq c_1 \sqrt{d^*/n_o}, \quad \mathbb{E}_{x \sim d_P^{\pi_e}}[\sqrt{k_{n_o}(x,x)}] \leq c_1 \sqrt{\frac{C^{\pi_e} d^* \{d^* + \log(c_2/\delta)\}}{n_o}}.$$

3. *Assume $\zeta^2 = \Omega(1)$. With probability $1 - \delta$,*

$$V_{P,c}^{\hat{\pi}_{\mathrm{IL}}} - V_{P,c}^{\pi_e} \leq \mathrm{Err}_o + \mathrm{Err}_e \tag{9}$$

$$\mathrm{Err}_o = c_1 H^2 \{d^* + \log(c_2/\delta)\} d^* \sqrt{\frac{d_{\mathcal{S}} C^{\pi_e} \log^3(c_2 d_{\mathcal{S}} n_o/\delta) \log(1+n_o)}{n_o}}$$

$$\mathrm{Err}_e = 2H \sqrt{\log(2|\mathcal{F}|/\delta)/(2n_e)}.$$

4. *Assume $\zeta^2 = \Omega(1)$. For offline RL, with probability $1 - \delta$,*

$$V_{P,c}^{\hat{\pi}_{\mathrm{RL}}} - V_{P,c}^{\pi^*} \leq c_1 H^2 \{d^* + \log(c_2/\delta)\} d^* \sqrt{\frac{d_{\mathcal{S}} C^{\pi^*} \log^3(c_2 d_{\mathcal{S}} n_o/\delta) \log(1+n_o)}{n_o}},$$

*where $C^{\pi^*} = \sup_{\|x\|_2 \leq 1} \frac{x \Sigma_{\pi^*} x}{x \Sigma_\rho x}$.*

The final bound in (9) suggests that $\mathrm{Err}_o$ is $\tilde{O}(H^2 \{d^*\}^2 \sqrt{d_{\mathcal{S}} C^{\pi_e}/n_o})$. In other words, when $C^{\pi_e}$, $d^*$ are not so large and the offline sample size is large enough, $\mathrm{Err}_e$ dominates $\mathrm{Err}_e$ and the covariate shift problem in BC can be avoided since the horizon dependence is just $H$. Our bound is the natural extension of [Theorem 21] to possibly infinite dimensional models.

The first and second statements in [Theorem 25] are mainly proved in two steps: formulating $k_{n_o}(x,x)$ into the variational representation and utilizing the uniform law with localization. Note the critical radius can be upper-bounded more tightly than $O(\sqrt{d^*/n_o})$ depending on the kernels. Besides, $C^{\pi_e}$ can be replaced with a tighter quantity:

$$\max_{i \in \mathbb{N}} \mathbb{E}_{(s,a) \sim d_P^{\pi_e}}[\psi_i^2(s,a)].$$

Since $\mathbb{E}_{(s,a) \sim \rho}[\psi_i^2(s,a)] = 1$, this quantity also measure the difference of batch data and expert. This is less than $C^{\pi_e}$ noting that $\frac{x \Sigma_{\pi^*} x}{x \Sigma_\rho x} = \mathbb{E}_{(s,a) \sim d_P^{\pi_e}}[\psi_i^2(s,a)]$ when $x$ is a vector s.t. only $i$-th element is 1 and the other elements are 0. The third statement in [Theorem 25] is directly proved by combining the second statement in [Theorem 25] and [Theorem 24].

**Implication to offline RL**    The final statement in [Theorem 25] is the bound for the RL case. This is the first result showing the error bound for pessimistic offline RL with nonparametric models. As related literature, in model-free offline RL, [70; 21] obtained the finite-sample error bounds characterized by the critical radius for some minimax-type estimators called Modified RBM [5]. As we show in [Theorem 25], since the critical radius of an RKHS ball is upper-bounded by the effective dimension $d^*$, their bounds are also characterized by the effective dimension. On top of that, several papers derived the bounds under the general function approximation setting: FQI [24; 21; 45; 16], marginal weighting based estimators [69], DICE methods [84; 46], policy based methods [42; 43] and MABO [77]. Comparing to our result, all of their bounds depend on

$$\sup_{\pi \in \Pi} \sup_{(s,a)} \frac{d_P^\pi(s,a)}{\rho(s,a)} \quad \text{or} \quad \sup_{(s,a)} \frac{1}{\rho(s,a)}.$$

The pessimistic bonus allows us to obtain the bound only depending on $C^{\pi^*}$ but not the above constants. Besides, our $C^{\pi^*}$ in Theorem 25 is more refined quantity than the density ratios in the sense that it is defined in terms of the relative condition number. Note we can easily obtain the statements which replace $C^{\pi^*}$ in Theorem 25 with $\frac{d_P^{\pi^*}(s,a)}{\rho(s,a)}$.

**Remark 26** (Relation with more general offline RL literature)**.** *Due to the lack of exploration, it is known how to deal with the lack of the coverage of the offline data is a challenging problem [83; 74]. We use the penalty terms based on model-based RL. In the above, we explain how the penalty term in* `MILO` *(and its RL counterpart) is transferred to the final sample-error bounds. The idea of penalization has been utilized in a variety of other ways in offline RL. The first other way is imposing constraints on the policy class or Q-function class so that estimated policies are not too much far away from behavior policies. For example, we can use KL divegences, MMD distance, Wasserstein distance to measure the distance from behavior policies [76; 23; 44; 68; 26] and add $D(\pi, \pi_b)$ as penalty terms, where $\pi_b$ is a behavior policy. Another way is explicitly estimating the lower bound of q-functions [39; 81; 82]. By doing so, we can avoid the overestimation of the q-functions in unknown (non-covered) regions.*

**Remark 27** (Relation with GP literature)**.** *The quantity $\mathbb{E}_{x \sim \rho(x)}[k_{n_o}(x, x)]$ is often referred to as the learning curve in GP literature [75; 61; 55]. Their analysis mainly focuses on the numerical viewpoints, that is, how to approximately calculate $\mathbb{E}_{x \sim \rho(x)}[k_{n_o}(x, x)]$. Though [41] analyzes the convergence property, their analysis is limited to the expectation and the result is asymptotic. As far as we know, our result is the first result showing the finite-sample error rate.*

**Remark 28** (Duality between KNRs and GPs)**.** *KNRs and GPs have a primal and dual relationship via Mercer's theorem. In fact, as we see, $k(\cdot, \cdot) = \langle \phi(\cdot), \phi(\cdot) \rangle$, we have $k_{n_o}(x, x) = \phi(x)^\top \Sigma_{n_o}^{-1} \phi(x)$. Thus, our result in GPs can be applied to the result for infinite-dimensional KNRs with $\phi : \mathcal{S} \times \mathcal{A} \mapsto \mathcal{H}$ where $\mathcal{H}$ is some RKHS.*

**Remark 29** (Online RL using RKHS)**.** *There are several online RL literature using RKHS such as the model-based way [13] like our work and the model-free way [3; 78; 19]. In both cases, the final-sample error bounds incur the maximum information gain, i.e., a worse case quantity which is distribution independent. Comparing to that, our final bounds use distribution-dependent quantities $d^*$.*

## C.4 Missing Proofs

Below, we provide missing proofs for tabular MDPs, KNRs, and non-parametric GP models.

### C.4.1 Missing proofs for tabular result

We start by providing proof of the tabular MDP result.

*Proof of Theorem 19.* We use Theorem 3. Then, we have

$$V_{P,c}^{\hat{\pi}_{\mathrm{IL}}} - V_{P,c}^{\pi_e} \leq (6H^2 + 2H) \min(1, \mathbb{E}_{(s,a) \sim d_P^{\pi_e}}[\sigma(s,a)]) + H\epsilon_{\mathrm{stat}}.$$

Hereafter, we show how to upper-bound $\mathbb{E}_{(s,a) \sim d_P^{\pi_e}}[\sigma(s,a)]$. We use Lemma 35. Then, by letting $\xi = c_1 \log(|\mathcal{S}||\mathcal{A}|c_2/\delta)$, with probability $1 - \delta$, we have

$$\frac{1}{N(s,a) + \lambda} \leq \frac{\xi}{n_o \rho(s,a) + \lambda} \quad \forall (s,a) \in \mathcal{S} \times \mathcal{A}.$$

We condition on the above event. Then,

$$\mathbb{E}_{(s,a) \sim d_P^{\pi_e}}[\sigma(s,a)] \leq \mathbb{E}_{(s,a) \sim d_P^{\pi_e}} \left[ \sqrt{\frac{|\mathcal{S}| \log 2 + \log(2|\mathcal{S}||\mathcal{A}|/\delta)}{2\{N(s,a) + \lambda\}}} + \frac{\lambda}{N(s,a) + \lambda} \right]$$

$$\leq \sqrt{\mathbb{E}_{(s,a) \sim d_P^{\pi_e}} \left[ \frac{|\mathcal{S}| \log 2 + \log(2|\mathcal{S}||\mathcal{A}|/\delta)}{2\{N(s,a) + \lambda\}} \right]} + \mathbb{E}_{(s,a) \sim d_P^{\pi_e}} \left[ \frac{\lambda}{N(s,a) + \lambda} \right].$$

From Lemma 35, we have

$$\mathbb{E}_{(s,a)\sim d_P^{\pi_e}}[\sigma(s,a)] \leq \sqrt{\xi\mathbb{E}_{(s,a)\sim d_P^{\pi_e}}\left[\frac{|\mathcal{S}|\log 2 + \log(2|\mathcal{S}||\mathcal{A}|/\delta)}{\{n_o\rho(s,a)+\lambda\}}\right]} + \mathbb{E}_{(s,a)\sim d_P^{\pi_e}}\left[\frac{\lambda\xi}{n_o\rho(s,a)+\lambda}\right]$$

$$\leq \sqrt{\xi C^{\pi_e}\mathbb{E}_{(s,a)\sim\rho}\left[\frac{|\mathcal{S}|\log 2 + \log(2|\mathcal{S}||\mathcal{A}|/\delta)}{\{n_o\rho(s,a)+\lambda\}}\right]} + C^{\pi_e}\mathbb{E}_{(s,a)\sim\rho}\left[\frac{\lambda\xi}{n_o\rho(s,a)+\lambda}\right]$$

$$\leq \sqrt{\xi C^{\pi_e}\sum_{s,a}\left[\frac{\{|\mathcal{S}|\log 2 + \log(2|\mathcal{S}||\mathcal{A}|/\delta)\}\rho(s,a)}{\{n_o\rho(s,a)+\lambda\}}\right]} + C^{\pi_e}\sum_{s,a}\left[\frac{\rho(s,a)\lambda\xi}{n_o\rho(s,a)+\lambda}\right]$$

$$\leq \sqrt{\xi C^{\pi_e}\{|\mathcal{S}|\log 2 + \log(2|\mathcal{S}||\mathcal{A}|/\delta)\}|S||A|/n_o} + \lambda C^{\pi_e}\xi|S||A|/n_o.$$

where again

$$C^{\pi_e} = \max_{(s,a)}\frac{d_P^{\pi_e}(s,a)}{\rho(s,a)}.$$

This concludes the proof. $\qquad\square$

### C.4.2 Missing proofs for KNR results

Next we move to provide proofs for the KNR results.

*Proof of Theorem 20.* In the proof, we use two statements, eq. (11) and eq. (12), in the proof of theorem 21. We recommend readers to read the proof of theorem 21 first.

We denote the eigenvalues of $\sum_{i=1}^{n_o}\phi(s_i,a_i)\phi^\top(s_i,a_i)$ by $\{\hat{\mu}_i\}_{i=1}^d$ s.t. $\hat{\mu}_1 \geq \hat{\mu}_2 \geq \cdots$. Since we assume $\|\phi(s,a)\|_2 \leq 1$, we have $\hat{\mu}_1 \leq n_o$.

**First step** We first show

$$\log(\det(\Sigma_{n_o})/\det(\lambda\mathbf{I})) \leq \mathrm{tr}\left[\Sigma_{n_o}^{-1}\sum_{i=1}^{n_o}\phi(s_i,a_i)\phi^\top(s_i,a_i)\right]\{\log(1+n_o/\lambda)+1\}.$$

Note this directly shows $\log(\det(\Sigma_{n_o})/\det(\lambda\mathbf{I})) \leq d\log(1+n_o/\lambda), \phi(s,a) \in \mathbb{R}^d$. The above is proved as follows:

$$\log(\det(\Sigma_{n_o})/\det(\lambda\mathbf{I})) = \sum_{i=1}^d\log\left(1+\frac{\hat{\mu}_i}{\lambda}\right) = \sum_{i=1}^d\log\left(1+\frac{\hat{\mu}_i}{\lambda}\right)\frac{\hat{\mu}_i/\lambda+1}{\hat{\mu}_i/\lambda+1}$$

$$= \sum_{i=1}^d\log\left(1+\frac{\hat{\mu}_i}{\lambda}\right)\frac{\hat{\mu}_i/\lambda}{\hat{\mu}_i/\lambda+1} + \log\left(1+\frac{\hat{\mu}_i}{\lambda}\right)\frac{1}{\hat{\mu}_i/\lambda+1}$$

$$\leq \log\left(1+\frac{\hat{\mu}_1}{\lambda}\right)\sum_{i=1}^d\frac{\hat{\mu}_i/\lambda}{\hat{\mu}_i/\lambda+1} + \sum_{i=1}^d\frac{\hat{\mu}_i/\lambda}{\hat{\mu}_i/\lambda+1} \qquad (\log(1+x) < x)$$

$$\leq \{\log(1+n_o/\lambda)+1\}\sum_{i=1}^d\frac{\hat{\mu}_i/\lambda}{\hat{\mu}_i/\lambda+1} \qquad (\hat{\mu}_1 \leq n_o)$$

$$= \{\log(1+n_o/\lambda)+1\}\mathrm{tr}\left[\Sigma_{n_o}^{-1}\sum_{i=1}^{n_o}\phi(s_i,a_i)\phi^\top(s_i,a_i)\right].$$

In the last line, letting $UVU^\top$ be the eigendecomopsition of $\sum_{i=1}^{n_o}\phi(s_i,a_i)\phi^\top(s_i,a_i)$, we use

$$\mathrm{tr}\left[\Sigma_{n_o}^{-1}\sum_{i=1}^{n_o}\phi(s_i,a_i)\phi^\top(s_i,a_i)\right] = \mathrm{tr}\left[\{V+\lambda\mathbf{I}\}^{-1}V\right] = \sum_{i=1}^d\frac{\hat{\mu}_i/\lambda}{\hat{\mu}_i/\lambda+1}.$$

Then, the first statement is proved.

**Second step** Next, we prove the second statement. We have

$$\mathrm{tr}\left[\Sigma_{n_o}^{-1}\sum_{i=1}^{n_o}\phi(s_i,a_i)\phi^\top(s_i,a_i)\right]=\sum_{i=1}^{n_o}\phi^\top(s_i,a_i)\Sigma_{n_o}^{-1}\phi(s_i,a_i).$$

Then, from (11), with probability $1-\delta$,

$$\sum_{i=1}^{n_o}\phi^\top(s_i,a_i)\Sigma_{n_o}^{-1}\phi(s_i,a_i)\lesssim c_1\{\mathrm{rank}[\Sigma_\rho]+\log(c_2/\delta)\}\sum_{i=1}^{n_o}\phi^\top(s_i,a_i)\{n_o\Sigma_\rho+\lambda\mathbf{I}\}^{-1}\phi(s_i,a_i).$$

$$(10)$$

Hereafter, we condition on the above event. To upper-bound $\sum_{i=1}^{n_o}\|\phi(s_i,a_i)\|^2_{\{n_o\Sigma_\rho+\lambda\mathbf{I}\}^{-1}}$, we use Bernstein's inequality:

$$\left|\sum_{i=1}^{n_o}\phi^\top(s_i,a_i)\{n_o\Sigma_\rho+\lambda\mathbf{I}\}^{-1}\phi(s_i,a_i)-n_o\mathbb{E}_{(s,a)\sim\rho}[\phi^\top(s,a)\{n_o\Sigma_\rho+\lambda\mathbf{I}\}^{-1}\phi(s,a)]\right|$$

$$\lesssim\sqrt{n_o\mathop{\mathrm{Var}}_{(s,a)\sim\rho}[\phi^\top(s,a)\{n_o\Sigma_\rho+\lambda\mathbf{I}\}^{-1}\phi(s,a)]}+1/\lambda.$$

since $\|\phi(s,a)\|^2_{\{n_o\Sigma_\rho+\lambda\mathbf{I}\}^{-1}}\leq 1/\lambda\,\forall(s,a)\in\mathcal{S}\times\mathcal{A}$. Here, from (12),

$$n_o\mathbb{E}_{(s,a)\sim\rho}[\phi^\top(s,a)\{n_o\Sigma_\rho+\lambda\mathbf{I}\}^{-1}\phi(s,a)]\leq\mathrm{rank}[\Sigma_\rho].$$

Besides,

$$\mathop{\mathrm{Var}}_{(s,a)\sim\rho}[\phi^\top(s,a)\{n_o\Sigma_\rho+\lambda\mathbf{I}\}^{-1}\phi(s,a)]\leq\mathbb{E}_{(s,a)\sim\rho}[\{\phi^\top(s,a)\{n_o\Sigma_\rho+\lambda\mathbf{I}\}^{-1}\phi(s,a)\}^2]$$

$$\leq 1/\lambda\mathbb{E}_{(s,a)\sim\rho}[\phi^\top(s,a)\{n_o\Sigma_\rho+\lambda\mathbf{I}\}^{-1}\phi(s,a)]$$

$$\leq\mathrm{rank}[\Sigma_\rho]/(n_o\lambda).\qquad\text{(from (12))}$$

Thus,

$$\sum_{i=1}^{n_o}\phi^\top(s_i,a_i)\{n_o\Sigma_\rho+\lambda\mathbf{I}\}^{-1}\phi(s_i,a_i)\lesssim\mathrm{rank}[\Sigma_\rho]+\sqrt{\mathrm{rank}[\Sigma_\rho]/\lambda}+1/\lambda.$$

By combining (10) with the above, we have

$$\log(\det(\Sigma_{n_o})/\det(\lambda\mathbf{I}))\leq c_1\mathrm{rank}(\Sigma_\rho)\{\mathrm{rank}(\Sigma_\rho)+\log(c_2/\delta)\}\log(1+n_oc_3).$$

from $\lambda=\Omega(1)$.

$$\square$$

Before proving Theorem 21, we first present some lemmas.

**Lemma 30** (Distribution change). *Consider two distributions $\rho_1\in\Delta(\mathcal{S}\times\mathcal{A})$ and $\rho_2\in\Delta(\mathcal{S}\times\mathcal{A})$, and a feature mapping $\phi:\mathcal{S}\times\mathcal{A}\mapsto\mathcal{H}$ where $\mathcal{H}$ is some Hilbert space (e.g., finite dimensional Euclidean space). Denote $C:=\sup_{x\in\mathcal{H}}\frac{x^\top\mathbb{E}_{s,a\sim\rho_1}\phi(s,a)\phi(s,a)^\top x}{x^\top\mathbb{E}_{s,a\sim\rho_2}\phi(s,a)\phi(s,a)^\top x}$. Then for any positive definition linear matrix ( operator $\Lambda$), we have:*

$$\mathbb{E}_{s,a\sim\rho_1}\phi(s,a)^\top\Lambda\phi(s,a)\leq C\mathbb{E}_{s,a\sim\rho_2}\phi(s,a)^\top\Lambda\phi(s,a).$$

*Proof.* Denote the eigendecomposition of $\Lambda=U\Sigma U^\top$ where $\{\sigma_i,u_i\}$ as the eigenvalue-eigenvector pairs. We have:

$$\mathbb{E}_{s,a\sim\rho_1}\phi(s,a)^\top\Lambda\phi(s,a)=\sum_{i=0}^\infty\sigma_iu_i^\top\mathbb{E}_{s,a\sim\rho_1}\phi(s,a)\phi(s,a)^\top u_i$$

$$\leq\sum_{i=0}^\infty\sigma_iCu_i^\top\mathbb{E}_{s,a\sim\rho_2}\phi(s,a)\phi(s,a)^\top u_i$$

$$=C\mathbb{E}_{s,a\sim\rho_2}\phi(s,a)^\top\Lambda\phi(s,a),$$

which concludes the proof. $\square$

*Proof of Theorem 21.* Here, we prove the first statement. We need to upper-bound

$$\mathbb{E}_{(s,a)\sim d_P^{\pi_e}}\left[\sqrt{\phi^\top(s,a)\Sigma_{n_o}^{-1}\phi(s,a)}\right].$$

As the first step, we use Jensen's inequality:

$$\mathbb{E}_{(s,a)\sim d_P^{\pi_e}}\left[\sqrt{\phi^\top(s,a)\Sigma_{n_o}^{-1}\phi(s,a)}\right] \leq \sqrt{\mathbb{E}_{(s,a)\sim d_P^{\pi_e}}\left[\phi^\top(s,a)\Sigma_{n_o}^{-1}\phi(s,a)\right]}.$$

Hereafter, we analyze $\mathbb{E}_{(s,a)\sim d_P^{\pi_e}}\left[\phi^\top(s,a)\Sigma_{n_o}^{-1}\phi(s,a)\right]$.

We first use the definition of the relative condition number $C^{\pi_e}$ and Lemma 30 to change distribution from $d_P^{\pi_e}$ to $\rho$, i.e., via Lemma 30, we have:

$$\mathbb{E}_{s,a\sim d_P^{\pi_e}}\phi(s,a)^\top\Sigma_{n_o}^{-1}\phi(s,a) \leq C^{\pi_e}\mathbb{E}_{s,a\sim\rho}\phi(s,a)^\top\Sigma_{n_o}^{-1}\phi(s,a).$$

Thus, below we just need to bound $\mathbb{E}_{s,a\sim\rho}\phi(s,a)^\top\Sigma_{n_o}^{-1}\phi(s,a)$.

**Concentration argument** In this step, we consider how to bound $\mathbb{E}_{(s,a)\sim\rho}[\phi^\top(s,a)\Sigma_{n_o}^{-1}\phi(s,a)]$. To do that, we show with probability $1-\delta$,

$$\phi^\top(s,a)\Sigma_{n_o}^{-1}\phi(s,a) \leq c_1\{\text{rank}(\Sigma_\rho) + \log(c_2/\delta)\}\phi^\top(s,a)\{n_o\Sigma_\rho + \lambda\mathbf{I}\}^{-1}\phi(s,a) \quad \forall(s,a)\in\mathcal{S}\times\mathcal{A} \tag{11}$$

We use the variational representation:

$$\phi^\top(s,a)\Sigma_{n_o}^{-1}\phi(s,a) = \sup_{\{a\in\mathbb{R}^d:a^\top\Sigma_{n_o}a\leq 1\}}\{a^\top\phi(s,a)\}^2$$
$$= \sup_{\{a\in\mathbb{R}^d:a^\top\Sigma_{n_o}a\leq 1,\|a\|_2^2\leq(1+\lambda)/\lambda,\|a^\top\phi\|_\infty\leq 1/\lambda\}}\{a^\top\phi(s,a)\}^2.$$

Note that in the first line, we use

$$\sup_{\{a\in\mathbb{R}^d:a^\top\Sigma_{n_o}a\leq 1\}}a^\top\phi(s,a) = \sup_{\{b\in\mathbb{R}^d:b^\top b\leq 1\}}b^\top\Sigma_{n_o}^{-1/2}\phi(s,a) = \|\phi(s,a)\|_{\Sigma_{n_o}^{-1}}.$$

From the first line to the second line, we use the fact that the maximization regarding $a$ is taken when $\tilde{a} = \Sigma_{n_o}^{-1}\phi(s,a)/\|\phi(s,a)\|_{\Sigma_{n_o}^{-1}}$ and

$$\|\tilde{a}\|_2^2 = \phi^\top(s,a)\Sigma_{n_o}^{-2}\phi(s,a)/\{\phi^\top(s,a)\Sigma_{n_o}^{-1}\phi(s,a)\} = (n_o+\lambda)/\lambda^2,$$
$$|\tilde{a}^\top\phi| \leq \|\phi(s,a)\|_{\Sigma_{n_o}^{-1}} \leq 1/\lambda \quad \forall(s,a)\in\mathcal{S}\times\mathcal{A},$$

noting $\|\phi(s,a)\|_2 \leq 1$. By defining $\bar{c} = (n_o+\lambda)/\lambda^2$, we have $\forall(s,a)\in\mathcal{S}\times\mathcal{A}$,

$$\phi^\top(s,a)\Sigma_{n_o}^{-1}\phi(s,a) = \sup_{\{a\in\mathbb{R}^d:a^\top\Sigma_{n_o}a\leq 1,\|a\|_2^2\leq\bar{c},\|a^\top\phi\|_\infty\leq 1/\lambda\}}\{a^\top\phi(s,a)\}^2$$
$$= \sup_{\{a\in\mathbb{R}^d:a^\top\lambda\mathbf{I}a+\sum_{i=1}^{n_o}\{a^\top\phi_i\}^2\leq 1,\|a\|_2^2\leq\bar{c},\|a^\top\phi\|_\infty\leq 1/\lambda\}}\{a^\top\phi(s,a)\}^2.$$

Next, we use Lemma 36, that is, with probability $1-\delta$,

$$\frac{1}{n_o}\sum_{i=1}^{n_o}f^2(s_i,a_i) \geq 0.5\mathbb{E}_{(s,a)\sim\rho}[f^2(s,a)] - 0.5\{\delta_{n_o}'\}^2 \,\forall f\in\mathcal{F},$$

where

$$\mathcal{F} = \{(s,a)\mapsto a^\top\phi(s,a):a^\top\Sigma_{n_o}a\leq 1,\|a\|_2^2\leq\bar{c},\|a^\top\phi\|_\infty\leq 1/\lambda,a\in\mathbb{R}^d\}.$$

Here, $\delta_{n_o}' = \delta_{n_o} + \sqrt{\log(c_2/\delta)/n_o}$, where $\delta_{n_o}$ is the critical radius of the function class $\mathcal{F}$. Noting $\lambda = \Omega(1)$, from Lemma 37, $\delta_{n_o}' = c_1\sqrt{\text{rank}[\Sigma_\rho]/n_o} + \sqrt{\log(c_2/\delta)/n_o}$. By conditioning on the

above event, $\forall (s,a) \in \mathcal{S} \times \mathcal{A}$, we have

$$\|\phi(s,a)\|_{\Sigma_{n_o}^{-1}}^2 \leq \sup_{\{a \in \mathbb{R}^d : a^\top \lambda \mathbf{I} a + 0.5 n_o \mathbb{E}_{(s,a) \sim \rho}[\{a^\top \phi\}^2] \leq 1 + 0.5 n_o \delta_{n_o}'^2, \|a\|_2^2 \leq \bar{c}, \|a^\top \phi\|_\infty \leq 1/\lambda\}} \{a^\top \phi(s,a)\}^2$$

$$\leq \sup_{\{a \in \mathbb{R}^d : a^\top \{n_o \Sigma_\rho + \lambda \mathbf{I}\} a \leq 2 + n_o \delta_{n_o}'^2, \|a\|_2^2 \leq \bar{c}, \|a^\top \phi\|_\infty < 1/\lambda\}} \{a^\top \phi(s,a)\}^2$$

$$\leq \sup_{\{a \in \mathbb{R}^d : a^\top \{n_o \Sigma_\rho + \lambda \mathbf{I}\} a \leq 2 + n_o \delta_{n_o}'^2\}} \{a^\top \phi(s,a)\}^2$$

$$= (2 + n_o \delta_{n_o}'^2) \phi^\top(s,a) \{n_o \Sigma_\rho + \lambda \mathbf{I}\}^{-1} \phi(s,a)$$

$$\leq c_1 \{\mathrm{rank}[\Sigma_\rho] + \log(c_2/\delta)\} \phi^\top(s,a) \{n_o \Sigma_\rho + \lambda \mathbf{I}\}^{-1} \phi(s,a).$$

**Last step** Then, the final bound is

$$\mathbb{E}_{(s,a) \sim d_P^{\pi_e}}[\|\phi(s,a)\|_{\Sigma_{n_o}^{-1}}] = \sqrt{C^{\pi_e} \mathbb{E}_{(s,a) \sim \rho}[\phi^\top(s,a) \Sigma_{n_o}^{-1} \phi(s,a)]}$$

$$\leq c_1 \sqrt{C^{\pi_e} n_o \{\mathrm{rank}[\Sigma_\rho] + \log(c_2/\delta)\} \mathbb{E}_{(s,a) \sim \rho}[\phi^\top(s,a) \{\Sigma_\rho + \lambda \mathbf{I}\}^{-1} \phi(s,a)]}.$$

Let $UVU^\top$ be the eigenvalue decomoposition of $\Sigma_\rho$ s.t. $V_{i,i} = \mu_i$. We have

$$\mathbb{E}_{(s,a) \sim \rho}[\phi^\top(s,a) \{n_o \Sigma_\rho + \lambda \mathbf{I}\}^{-1} \phi(s,a)] = \mathrm{Tr}[\{n_o \Sigma_\rho + \lambda \mathbf{I}\}^{-1} \{\Sigma_\rho\}] = \mathrm{Tr}[\{n_o V + \lambda \mathbf{I}\}^{-1} V]$$

$$= \frac{1}{n_o} \sum_{i=1}^{n_o} \frac{\mu_i}{\mu_i + \lambda/n_o} \leq \frac{\mathrm{rank}[\Sigma_\rho]}{n_o}. \tag{12}$$

By combining all things together, with probability $1 - \delta$,

$$\mathbb{E}_{(s,a) \sim d_P^{\pi_e}}[\|\phi(s,a)\|_{\Sigma_{n_o}^{-1}}] \leq c_1 \sqrt{\frac{C^{\pi_e} \mathrm{rank}[\Sigma_\rho] \{\mathrm{rank}[\Sigma_\rho] + \log(c_2/\delta)\}}{n_o}}.$$

$\square$

### C.4.3 Missing proofs of non-parametric model

Finally, we provide missing proofs for the non-parametric GP model.

*Proof of Theorem 24.* In the proof, we use two statements, (13) and (14), in the proof of Theorem 25. We recommend readers to read the proof of Theorem 25 first.

We denote the eigenvalues of $\mathbf{K}_{n_o}$ by $\{\hat{\mu}_i\}_{i=1}^{n_o}$ s.t. $\hat{\mu}_1 \geq \hat{\mu}_2 \geq \cdots$. From Assumption 22, we have

$$n_o = \mathrm{tr}(\mathbf{K}_{n_o}) = \sum_{i=1}^{n_o} \hat{\mu}_i.$$

Thus implies $\hat{\mu}_1 \leq n_o$. Then,

$$\log(\det(\mathbf{I} + \zeta^{-2} \mathbf{K}_{n_o})) = \sum_{i=1}^{n_o} \log\left(1 + \frac{\hat{\mu}_i}{\zeta^2}\right) = \sum_{i=1}^{n_o} \log\left(1 + \frac{\hat{\mu}_i}{\zeta^2}\right) \frac{\hat{\mu}_i/\zeta^2 + 1}{\hat{\mu}_i/\zeta^2 + 1}$$

$$= \sum_{i=1}^{n_o} \log\left(1 + \frac{\hat{\mu}_i}{\zeta^2}\right) \frac{\hat{\mu}_i/\zeta^2}{\hat{\mu}_i/\zeta^2 + 1} + \log\left(1 + \frac{\hat{\mu}_i}{\zeta^2}\right) \frac{1}{\hat{\mu}_i/\zeta^2 + 1}$$

$$= \sum_{i=1}^{n_o} \log\left(1 + \frac{\hat{\mu}_i}{\zeta^2}\right) \frac{\hat{\mu}_i/\zeta^2}{\hat{\mu}_i/\zeta^2 + 1} + \log\left(1 + \frac{\hat{\mu}_i}{\zeta^2}\right) \frac{1}{\hat{\mu}_i/\zeta^2 + 1}$$

$$\leq \log\left(1 + \frac{\hat{\mu}_1}{\zeta^2}\right) \sum_{i=1}^{n_o} \frac{\hat{\mu}_i/\zeta^2}{\hat{\mu}_i/\zeta^2 + 1} + \sum_{i=1}^{n_o} \frac{\hat{\mu}_i/\zeta^2}{\hat{\mu}_i/\zeta^2 + 1} \qquad (\log(1+x) \leq x)$$

$$\leq \{\log(1 + n_o/\zeta^2) + 1\} \sum_{i=1}^{n_o} \frac{\hat{\mu}_i/\zeta^2}{\hat{\mu}_i/\zeta^2 + 1} \qquad (\hat{\mu}_1 \leq n_o)$$

$$\leq \{\log(1 + n_o/\zeta^2) + 1\} \min_j \{j + \hat{B}(j+1)/\zeta^2\} \leq 2\{\log(1 + n_o/\zeta^2) + 1\} \hat{d},$$

where the last second inequality uses the fact that $\sum_{i=1}^{n_o} \frac{\hat{\mu}_i/\xi^2}{\hat{\mu}_i/\xi^2+1} \le j + \sum_{i=j+1}^{n_o} \hat{\mu}_i/\xi^2$. Then, the first statement is proved.

Next, we prove the second statement. We use
$$\sum_{i=1}^{n_o} \frac{\hat{\mu}_i/\zeta^2}{\hat{\mu}_i/\zeta^2+1} = \frac{1}{\zeta^2} \sum_{i=1}^{n_o} k_{n_o}(x_i, x_i).$$
proved in Lemma 40. Then, from (13), with probability $1 - \delta$,
$$\frac{1}{\zeta^2} \sum_{i=1}^{n_o} k_{n_o}(x_i, x_i) \lesssim \delta_{n_o}'^2 \sum_{i=1}^{n_o} \sup_{\{f:\zeta^2/n_o\|f\|_k^2+\mathbb{E}_{x\sim\rho}[f^2(x)]\le 1, f\in\mathcal{H}_k\}} f^2(x_i),$$
where $\delta_n' = \delta_n + \sqrt{\log(c_2/\delta)/n_o}$ and $\delta_n$ is the critical radius of $\{f \in \mathcal{H}_k : \|f\|_k \le 1\}$. Hereafter, we condition on the above event.

Then, from Bernstein's inequality,
$$\left| \left\{ \sum_{i=1}^{n_o} \sup_{\{f:\zeta^2/n_o\|f\|_k^2+\mathbb{E}_{x\sim\rho}[f^2(x)]\le 1, f\in\mathcal{H}_k\}} f^2(x_i) \right\} - n_o \mathbb{E}_{x\sim\rho} \left[ \sup_{\{f:\zeta^2/n_o\|f\|_k^2+\mathbb{E}_{x\sim\rho}[f^2(x)]\le 1, f\in\mathcal{H}_k\}} f^2(x) \right] \right|$$
$$\lesssim \sqrt{n_o \operatorname{Var}_{x\sim\rho}[\sup_{\{f:\zeta^2/n_o\|f\|_k^2+\mathbb{E}_{x\sim\rho}[f^2(x)]\le 1, f\in\mathcal{H}_k\}} f^2(x)]} + n_o.$$

We use for $f$ in $\mathcal{H}_k$ s.t. $\|f\|_k \le 1$
$$|f(x)| = |\langle f(\cdot), k(x, \cdot)\rangle_k| \le \|f\|_k \|k(x, \cdot)\|_k \le 1.$$
from assumption 22. Here, from (14), the expectation is upper-bounded by
$$\mathbb{E}_{x\sim\rho} \left[ \sup_{\{f:\zeta^2/n_o\|f\|_k^2+\mathbb{E}_{x\sim\rho}[f^2(x)]\le 1, f\in\mathcal{H}_k\}} f^2(x) \right] \le d^*.$$
Besides, the variance is also upper-bounded by
$$\operatorname{Var}_{x\sim\rho}[\sup_{\{f:\zeta^2/n_o\|f\|_k^2+\mathbb{E}_{x\sim\rho}[f^2(x)]\le 1, f\in\mathcal{H}_k\}} f^2(x)]$$
$$\le \mathbb{E}_{x\sim\rho}[\sup_{\{f:\zeta^2/n_o\|f\|_k^2+\mathbb{E}_{x\sim\rho}[f^2(x)]\le 1, f\in\mathcal{H}_k\}} f^4(x)]$$
$$\le \mathbb{E}_{x\sim\rho}[\sup_{\{f:\zeta^2/n_o\|f\|_k^2+\mathbb{E}_{x\sim\rho}[f^2(x)]\le 1, f\in\mathcal{H}_k\}} f^2(x)]$$
$$(f^2(x) \le 1 \,\forall x \in \mathcal{S} \times \mathcal{A} \text{ from Assumption 22})$$
$$= d^*. \qquad\qquad\qquad (\text{From (14)})$$
Thus, with probability $1 - \delta$,
$$\sum_{i=1}^{n_o} k_{n_o}(x_i, x_i) \lesssim \{\delta_{n_o}'\}^2 n_o (d^* + \sqrt{d^*} + 1)$$
$$\lesssim c_1 \{d^* + \log(c_2/\delta)\} d^*.$$
noting $\delta_{n_o}' = \sqrt{d^*/n_o} + \sqrt{\log(c_2/\delta)/n_o}$ from Theorem 25.

By combining all things together, with probability $1 - \delta$,
$$\log(\det(\mathbf{I} + \zeta^{-2}\mathbf{K}_{n_o})) \le \{\log(1 + n_o/\zeta^2) + 1\} \sum_{i=1}^{n_o} \frac{\hat{\mu}_i/\zeta^2}{\hat{\mu}_i/\zeta^2+1}$$
$$= \{\log(1 + n_o/\zeta^2) + 1\} \frac{1}{\zeta^2} \sum_{i=1}^{n_o} k_{n_o}(x_i, x_i)$$
$$\lesssim \{\log(1 + c_3 n_o)\}\{d^* + \log(c_2/\delta)\} d^*.$$
This concludes the proof.

$\square$

*Proof of Theorem 25.*

**First Statement** From Jensen's inequality, we have

$$\mathbb{E}_{x \sim d_P^{\pi_e}}\left[\sqrt{k_{n_o}(x,x)}\right] \le \sqrt{\mathbb{E}_{x \sim d_P^{\pi_e}}[k_{n_o}(x,x)]}.$$

Thus, we focus how to bound $\mathbb{E}_{x \sim d_P^{\pi_e}}[k_{n_o}(x,x)]$. Before that, we show the following statement. With probability $1 - \delta$, we have for $\forall x \in \mathcal{S} \times \mathcal{A}$:

$$k_{n_o}(x,x) \le c_1 \zeta^2 \delta_{n_o}'^2 \times \sup_{\{f \in \mathcal{H}_k : \zeta^2/n_o \|f\|_k^2 + \mathbb{E}_{x \sim \rho}[f^2(x)] \le 1\}} f^2(x), \tag{13}$$

where $\delta_{n_o}' = \delta_{n_o} + \sqrt{\log(c_2/\delta)/n_o}$ and $\delta_{n_o}$ is the critical radius of $\{f \in \mathcal{H}_k : \|f\|_k \le 1\}$.

As the first step, we use Lemma 38 and Lemma 39.

$$k_{n_o}(x,x) = \sup_{\{f \in \mathcal{H}_{k_{n_o}} \| \|f\|_{k_{n_o}}^2 \le 1\}} f^2(x) \qquad \text{(From Lemma 38)}$$

$$= \sup_{\{f \in \mathcal{H}_k \| \|f\|_k^2 + \zeta^{-2} \sum_{i=1}^{n_o} f(x_i)^2 \le 1\}} f^2(x). \qquad \text{(From Lemma 39)}$$

Next invoke Lemma 36, that is, with probability $1 - \delta$,

$$\frac{1}{n_o} \sum_{i=1}^{n_o} f^2(x_i) \ge 0.5 \mathbb{E}_{(s,a) \sim \rho}[f^2(x)] - 0.5 \{\delta_{n_o}'\}^2 \ \forall f \in \mathcal{F}$$

where

$$\mathcal{F} = \{f : f \in \mathcal{H}_k, \|f\|_k^2 = 1\}.$$

Here, $\delta_{n_o}' = \delta_{n_o} + \sqrt{\log(c_2/\delta)/n_o}$, where $\delta_{n_o}$ is the critical radius of the function class $\mathcal{F}$. Hereafter, we condition on the above event. Note the uniform boundedness assumption of $\mathcal{F}$ for Lemma 36 is satisfied noting

$$|f(x)| = |\langle f(\cdot), k(\cdot, x)\rangle_k| \le \|f\|_k \|k(\cdot, x)\|_k \le 1.$$

noting assumption 22. Then, we have

$$k_{n_o}(x,x) \le \sup_{\{f \in \mathcal{H}_k \| \|f\|_k^2 + \zeta^{-2} n_o/2 \mathbb{E}_{x \sim \rho}[f^2(x)] \le 1 + n_o \delta_{n_o}'^2/2\}} f^2(x).$$

$k_{n_o}(x,x)$ is further upper-bounded by

$$k_{n_o}(x,x) \le \sup_{\{f \in \mathcal{H}_k : 2\zeta^2/n_o \|f\|_k^2 + \mathbb{E}_{x \sim \rho}[f^2(x)] \le 2\zeta^2/n_o + \zeta^2 \delta_{n_o}'^2\}} f^2(x) \qquad \text{(Multiply } 2\zeta^2/n_o)$$

$$\le (2\zeta^2/n_o + \zeta^2 \delta_{n_o}'^2) \times \sup_{\{f \in \mathcal{H}_k : \zeta^2/n_o \|f\|_k^2 + \mathbb{E}_{x \sim \rho}[f^2(x)] \le 1\}} f^2(x)$$

$$\le c_1 \zeta^2 \delta_{n_o}'^2 \times \sup_{\{f \in \mathcal{H}_k : \zeta^2/n_o \|f\|_k^2 + \mathbb{E}_{x \sim \rho}[f^2(x)] \le 1\}} f^2(x).$$

This concludes (13).

Next, we show

$$\mathbb{E}_{x \sim d_P^{\pi_e}}\left[\sup_{\{f \in \mathcal{H}_k : \zeta^2/n_o \|f\|_k^2 + \mathbb{E}_{x \sim \rho}[f^2(x)] \le 1\}} f^2(x)\right] \le 2d^* \times \sup_{\|x\|_2 \le 1} \frac{x^\top \Sigma_{\pi_e} x}{x^\top \Sigma_\rho x}.$$

For $f(\cdot) = a^\top \phi(\cdot)$ (recall $\phi(\cdot)$ is the feature mapping defined by the eigenvalues $\mu_i$ and eigenfunctions $\phi$, s.t. $\phi = (\phi_1, \cdots, \phi_\infty)$), we have

$$\|f\|_k^2 = a^\top a, \quad \mathbb{E}_{x \sim \rho}[f^2(x)] = a^\top M a.$$

where $M$ is a diagonal matrix in $\mathbb{R}^{\infty \times \infty}$ s.t. $M_{i,i} = \mu_i$. Thus,

$$\mathbb{E}_{x \sim d_P^{\pi_e}}\left[\sup_{\{f : \zeta^2/n_o \|f\|_k^2 + \mathbb{E}_{x \sim \rho}[f^2(x)] \le 1, f \in \mathcal{H}_k\}} f^2(x)\right] = \mathbb{E}_{x \sim d_P^{\pi_e}}\left[\sup_{\{a \in \mathbb{R}^\infty : a^\top(\zeta^2/n_o \mathbf{I} + M)a \le 1\}} \{a^\top \phi(x)\}^2\right].$$

Then, by letting $\Sigma_\rho$ and $\Sigma_{\pi_e}$ be $\mathbb{E}_{(s,a)\sim\rho}[\phi(s,a)\phi^\top(s,a)]$ and $\mathbb{E}_{(s,a)\sim d_P^{\pi_e}}[\phi(s,a)\phi^\top(s,a)]$,

$$
\begin{aligned}
\mathbb{E}_{x\sim d_P^{\pi_e}}\left[\sup_{\{a\in\mathbb{R}^d:a^\top(\zeta^2/n_o\mathbf{I}+M)a\leq 1\}}\{a^\top\phi(x)\}^2\right] &= \mathbb{E}_{x\sim d_P^{\pi_e}}[\phi(x)\{\zeta^2/n_o\mathbf{I}+M\}^{-1}\phi(x)] \\
&= \mathrm{tr}[\mathbb{E}_{x\sim d_P^{\pi_e}}[\phi(x)\phi(x)^\top]\{\zeta^2/n_o\mathbf{I}+M\}^{-1}] \\
&= \mathrm{tr}[\mathbb{E}_{x\sim\rho}[\phi(x)\phi(x)^\top]\{\zeta^2/n_o\mathbf{I}+M\}^{-1}]\times\sup_{\|x\|_2\leq 1}\frac{x^\top\Sigma_{\pi_e}x}{x^\top\Sigma_\rho x} \\
&= \sum_{i=1}^\infty\frac{\mu_i}{\zeta^2/n_o+\mu_i}\times\sup_{\|x\|_2\leq 1}\frac{x^\top\Sigma_{\pi_e}x}{x^\top\Sigma_\rho x}.
\end{aligned}
$$

Then, by defining $C^{\pi_e}=\sup_{\|x\|_2\leq 1}\frac{x^\top\Sigma_{\pi_e}x}{x^\top\Sigma_\rho x}$, we have

$$
\begin{aligned}
\mathbb{E}_{x\sim d_P^{\pi_e}}\left[\sup_{\{a\in\mathbb{R}^d:a^\top(\zeta^2/n_o+M)a\leq 1\}}\{a^\top\phi(x)\}^2\right] &\leq \min_j\{j+n_o/\zeta^2\sum_{i=j+1}^\infty\mu_i\}\times C^{\pi_e} \\
&\leq \min_j\{j+n_o/\zeta^2\sum_{i=j+1}^\infty\mu_i\}\times C^{\pi_e} \\
&\leq 2d^*\times C^{\pi_e}.
\end{aligned}
$$

By combining all things together ([(13)](#) and [(14)](#)), the statement is concluded, that is, with probability $1-\delta$:

$$
\begin{aligned}
\mathbb{E}_{d_P^{\pi_e}}[\sqrt{k_{n_o}(x,x)}] &\leq \sqrt{\zeta^2\delta_{n_o}'^2\times\mathbb{E}_{x\sim d_P^\pi}[\sup_{\{f\in\mathcal{H}_k:\zeta^2/n_o\|f\|_k^2+\mathbb{E}_{x\sim\rho}[f^2(x)]\leq 1\}}f^2(x)]} \\
&\leq \zeta\delta_{n_o}'\sqrt{C^{\pi_e}d^*}.
\end{aligned}
$$

where $C^{\pi_e}=\sup_{\|x\|_2\leq 1}\frac{x^\top\Sigma_{\pi_e}x}{x^\top\Sigma_\rho x}$.

**Remark 31.** *Like the above, We can also prove*

$$
\mathbb{E}_{x\sim\rho}[\sup_{\{f\in\mathcal{H}_k:\zeta^2/n_o\|f\|_k^2+\mathbb{E}_{x\sim\rho}[f^2(x)]\leq 1\}}f^2(x)]\leq\sum_{i=1}^\infty\frac{\mu_i}{\zeta^2/n_o+\mu_i}\leq 2d^*. \tag{14}
$$

*This is used in the proof of theorem [24](#).*

**Remark 32.** *We can also use*

$$
\begin{aligned}
\mathbb{E}_{x\sim d_P^{\pi_e}}\left[\sup_{\{a\in\mathbb{R}^d:a^\top(\zeta^2/n_o\mathbf{I}+M)a\leq 1\}}\{a^\top\phi(x)\}^2\right] &= \mathbb{E}_{x\sim d_P^{\pi_e}}[\phi(x)\{\zeta^2/n_o\mathbf{I}+M\}^{-1}\phi(x)] \\
&= \mathrm{tr}[\mathbb{E}_{x\sim d_P^{\pi_e}}[\phi(x)\phi(x)^\top]\{\zeta^2/n_o\mathbf{I}+M\}^{-1}] \\
&= \sum_{i=1}^\infty\frac{\mathbb{E}_{x\sim d_P^{\pi_e}}[\phi_i(x)\phi_i(x)^\top]\}}{\zeta^2/n_o+\mu_i} \\
&= \sum_{i=1}^\infty\frac{\mu_i}{\zeta^2/n_o+\mu_i}\times\left\{\frac{\mathbb{E}_{x\sim d_P^{\pi_e}}[\phi_i(x)\phi_i(x)^\top]}{\mu_i}\right\} \\
&= \sum_{j=1}^\infty\frac{\mu_j}{\zeta^2/n_o+\mu_j}\times\max_i(\mathbb{E}_{x\sim d_P^{\pi_e}}[\psi_i(x)\psi_i(x)^\top]).
\end{aligned}
$$

*Then, $C^{\pi_e}$ is replaced with $\max_i(\mathbb{E}_{x\sim d_P^{\pi_e}}[\psi_i(x)\psi_i(x)^\top])$.*

**Second statement** We use [Lemma 41](#) to calculate the critical radius of the RKHS ball. The critical inequality is

$$\sqrt{1/n_o}\sqrt{\sum_{i=1}^{n_o}\min(y^2,\mu_j)} \le y^2.$$

We show $y = \sqrt{d^*/n_o}$ satisfies the above. This is proved by

$$
\begin{aligned}
\sqrt{1/n_o}\sqrt{\sum_{i=1}^{n_o}\min(y^2,\mu_j)} &\le \min_{1\le k\le n_o}\{\sqrt{1/n}\sqrt{ky^2 + B(k+1)}\} \\
&\le \sqrt{1/n_o}\sqrt{d^*y^2 + B(d^*+1)} && (d^* \le n_o) \\
&\le \sqrt{1/n_o}\sqrt{d^*y^2 + d^*/n_o} && (B(d^*+1)\le d^*/n_o) \\
&\le \sqrt{d^*y^2/n_o} \le y^2.
\end{aligned}
$$

$\square$

# D   Auxiliary Lemmas

**Lemma 33** (Simulation Lemma). *Consider any two functions $f : \mathcal{S}\times\mathcal{A}\mapsto [0,1]$ and $\widehat{f} : \mathcal{S}\times\mathcal{A}\mapsto [0,1]$, any two transitions $P$ and $\widehat{P}$, and any policy $\pi : \mathcal{S}\mapsto \Delta(\mathcal{A})$. We have:*

$$
\begin{aligned}
V^\pi_{P;f} - V^\pi_{\widehat{P},\widehat{f}} &= \sum_{h=0}^{H}\mathbb{E}_{s,a\sim d^\pi_P}\left[f(s,a)-\widehat{f}(s,a)+\mathbb{E}_{s'\sim P(\cdot|s,a)}[V^\pi_{\widehat{P},\widehat{f};h}(s')]-\mathbb{E}_{s'\sim\widehat{P}(\cdot|s,a)}[V^\pi_{\widehat{P},\widehat{f};h}(s')]\right] \\
&\le \sum_{h=0}^{H}\mathbb{E}_{s,a\sim d^\pi_P}\left[f(s,a)-\widehat{f}(s,a)+\|V^\pi_{\widehat{P},\widehat{f};h}\|_\infty\|P(\cdot|s,a)-\widehat{P}(\cdot|s,a)\|_1\right].
\end{aligned}
$$

*where $V^\pi_{P,f;h}$ denotes the value function at time step $h$, under $\pi, P, f$.*

Such simulation lemma is standard in model-based RL literature and the derivation can be found, for instance, in the proof of Lemma 10 from [64].

**Lemma 34** ($\ell_1$ Distance between two Gaussians). *Consider two Gaussian distributions $P_1 := \mathcal{N}(\mu_1,\zeta^2 I)$ and $P_2 := \mathcal{N}(\mu_2,\zeta^2 I)$. We have:*

$$\|P_1 - P_2\|_1 \le \frac{1}{\zeta}\|\mu_1 - \mu_2\|_2.$$

This lemma is proved by Pinsker's inequality and the closed-form of the KL divergence between $P_1$ and $P_2$.

**Lemma 35** (Concentration on the inverse of state-action visitation). *We set $\lambda = \Omega(1)$. Then, with probability $1-\delta$,*

$$\frac{1}{N(s,a)+\lambda} \le \frac{c_1\log(|\mathcal{S}||\mathcal{A}|c_2/\delta)}{n_o\rho(s,a)+\lambda} \quad \forall(s,a)\in\mathcal{S}\times\mathcal{A}.$$

The extension of this lemma to the linear models is stated in eq. ([11](#)).

*Proof.* We set $\xi = c_1\log(|\mathcal{S}||\mathcal{A}|/\delta)+1\,(c_1 > 4/3 + 3)$. First, we have

$$\frac{1}{N(s,a)+\lambda} \le \frac{\xi}{N(s,a)+\xi\lambda}.$$

from $\xi \ge 1$. Here, by Bernsteins's inequality, with probability $1-\delta$,

$$N(s,a) \ge n_o\rho(s,a) - 2\sqrt{2n_o\rho(s,a)(1-\rho(s,a))\log(|\mathcal{S}||\mathcal{A}|/\delta)} - 4\log(|\mathcal{S}||\mathcal{A}|/\delta)/3,\ \forall(s,a).$$

Thus, $\forall (s,a) \in \bar{V}$, we have

$$
\begin{aligned}
N(s,a) + \xi\lambda &\geq n_o\rho(s,a) - 2\sqrt{2n_o\rho(s,a)(1-\rho(s,a))\log(|\mathcal{S}||\mathcal{A}|/\delta)} - 4\log(|\mathcal{S}||\mathcal{A}|/\delta)/3 + \xi\lambda \\
&\geq n_o\rho(s,a) - 2\sqrt{2n_o\rho(s,a)(1-\rho(s,a))\log(|\mathcal{S}||\mathcal{A}|/\delta)} + (c_1 - 4/3)\log(|\mathcal{S}||\mathcal{A}|/\delta) + \lambda \\
&\geq n_o\rho(s,a) - 2\sqrt{2n_o\rho(s,a)\log(|\mathcal{S}||\mathcal{A}|/\delta)} + (c_1 - 4/3)\log(|\mathcal{S}||\mathcal{A}|/\delta) + \lambda \\
&\geq 0.5n_o\rho(s,a) + (\sqrt{0.5n_o\rho(s,a)} - \sqrt{4\log(|\mathcal{S}||\mathcal{A}|/\delta)})^2 + (c_1 - 4/3 - 4)\log(|\mathcal{S}||\mathcal{A}|/\delta) + \lambda \\
&\geq 0.5n_o\rho(s,a) + 0.5\lambda.
\end{aligned}
$$

This implies with $1 - \delta$,

$$
\frac{1}{N(s,a) + \lambda} \leq \frac{2\xi}{n_0\rho(s,a) + \lambda} \; \forall(s,a).
$$

Then, noting $c_1\log(|\mathcal{S}||\mathcal{A}|/\delta) + 1 \leq c_1\log(|\mathcal{S}||\mathcal{A}|c_2/\delta)$ for some $c_2$, the proof is concluded. $\qquad\square$

**Lemma 36** (A uniform law with localization: Theorem 14.1 in [73]). *Assume $\|\mathcal{F}\|_\infty \leq b$. Denote the critical radius of a function class $\mathcal{F}$ by $\delta_n$. The critical radius $\delta_n$ is defined as a solution to*

$$
\mathcal{R}_{n_o}(y; \mathcal{F}) \leq y^2/b.
$$

*w.r.t $y$. Then, with probability $1 - \delta$*

$$
\frac{1}{n_o}\sum_{i=1}^{n_o} f(x_i)^2 \geq 1/2\mathbb{E}_{x\sim\rho}[f^2(x)] - (\delta'_n)^2/2 \quad \forall f \in \mathcal{F},
$$

*where $\delta'_n = \delta_n + c_1\sqrt{\log(c_2/\delta)/n_o}$.*

**Lemma 37** (Critical radius of linear models). *Assume $\|\phi(s,a)\|_2 \leq 1$ for any $(s,a) \in \mathcal{S} \times \mathcal{A}$. Then, the critical radius of function class $\mathcal{F} = \{(s,a) \mapsto a^\top\phi(s,a) : \|a\|_2^2 \leq \alpha, a^\top\phi \leq \beta, a \in \mathbb{R}^d\}$ is upper-bounded by*

$$
c\sqrt{\beta\mathrm{rank}(\Sigma_\rho)/n_o}.
$$

*where $c$ is a universal constant.*

We follow the proof of [73, Chapter 14]. Their argument depends on the assumption $\Sigma_\rho$ is full rank. We need to change the proof so that the full-rank assumption is removed and the rank $\mathrm{rank}[\Sigma_\rho]$ would appear in the final bound instead of $d$. Note that the final bound does not include $\alpha$.

*Proof.* Unless otherwise noted, in this proof, $\mathbb{E}[\cdot]$ is taken w.r.t.

$$
x_i = (s_i, a_i) \sim \rho(s,a), \quad \epsilon_i \sim 2\{\mathrm{Ber}(0.5) - 1\}.
$$

Note that $x_i$ and $\epsilon_i$ are independent.

Noting $\mathbb{E}_{\rho\sim(s,a)}[(a^\top\phi(s,a))^2] = a^\top\Sigma_\rho a$, the localized Rademacher complexity of $\mathcal{F}$, $\mathcal{R}_{n_o}(\xi; \mathcal{F})$, is

$$
\mathbb{E}\left[\sup_{\{b\in\mathbb{R}^d:\|b\|_2^2\leq\alpha,\|b\|_{\Sigma_\rho}\leq\xi,b^\top\phi\leq\beta\}} \left|\frac{1}{n_o}\sum_{i=1}^{n_o}\epsilon_i\{b^\top\phi(s_i,a_i)\}\right|\right]
$$

where $\{\epsilon_i\}_{i=1}^{n_o}$ is a set of independent Rademacher variables. This is upper-bounded by

$$
\mathbb{E}\left[\sup_{\{b\in\mathbb{R}^d:\|b\|_2^2\leq\alpha,\|b\|_{\Sigma_\rho}\leq\xi\}} \left|\frac{1}{n_o}\epsilon^\top\Phi b\right|\right]
$$

where $\Phi$ is a $n_o \times d$ design matrix s.t. the $i$-th row is $\phi^\top(s_i, a_i)$ and $\epsilon = (\epsilon_1, \cdots, \epsilon_{n_o})^\top$.

Here, we have $\mathbb{E}[\Phi^\top\Phi] = n_o\Sigma_\rho$. Let $UVU^\top$ be the SVD of $\Sigma_\rho$, where $U$ is a $n \times \mathrm{rank}[\Sigma_\rho]$ matrix and $V$ is a $\mathrm{rank}[\Sigma_\rho] \times \mathrm{rank}[\Sigma_\rho]$ diagonal matrix. Noting $b = UU^\top b + (\mathbf{I} - UU^\top)b$, we have

$$
\mathbb{E}\left[\sup_{\{b\in\mathbb{R}^d:\|b\|_2^2\leq\alpha,\|b\|_{\Sigma_\rho}\leq\xi\}} |\frac{1}{n_o}\epsilon^\top\Phi\{UU^\top b + (\mathbf{I}-UU^\top)b\}|\right]
$$

$$
\leq \mathbb{E}\left[\sup_{\{b\in\mathbb{R}^d:\|b\|_2^2\leq\alpha\}} |\frac{1}{n_o}\epsilon^\top\Phi(\mathbf{I}-UU^\top)b\}|\right] + \mathbb{E}\left[\sup_{\|b\|_{\Sigma_\rho}\leq\xi} |\frac{1}{n_o}\epsilon^\top\Phi UU^\top b\}|\right]
$$

$$
\leq \mathbb{E}\left[\sup_{\{b\in\mathbb{R}^d:\|b\|_2^2\leq\alpha\}} |\frac{1}{n_o}\epsilon^\top\Phi(\mathbf{I}-UU^\top)b\}|\right] + \mathbb{E}\left[\sup_{\|c\|_V\leq\xi} |\frac{1}{n_o}\epsilon^\top\Phi Uc\}|\right] \qquad (U^\top c = b)
$$

$$
\leq \mathbb{E}\left[\frac{\alpha}{n_o}\|\epsilon^\top\Phi(\mathbf{I}-UU^\top)\}\|_2\right] + \frac{\zeta}{n_o}\mathbb{E}\left[\|\epsilon^\top\Phi U\}\|_{V^{-1}}\right] \qquad \text{(CS inequality)}
$$

$$
\leq \frac{\alpha}{n_o}\sqrt{\mathbb{E}\left[\|\epsilon^\top\Phi(\mathbf{I}-UU^\top)\}\|_2^2\right]} + \frac{\zeta}{n_o}\sqrt{\mathbb{E}\left[\|\epsilon^\top\Phi U\|_{V^{-1}}^2\right]}. \qquad \text{(Jensen's inequality)}
$$

We analyze the second term and first term respectively.

Regarding the second term, we have

$$
\mathbb{E}_\epsilon[\|\epsilon^\top\Phi U\|_{V^{-1}}^2] = \mathbb{E}_\epsilon[\epsilon^\top\Phi U V^{-1}U^\top\Phi^\top\epsilon] = \mathrm{tr}(\Phi U V^{-1}U^\top\Phi^\top),
$$

where $\mathbb{E}_\epsilon[\cdot]$ is an expectation only regarding $\epsilon$. Then, by the law of total expectation,

$$
\begin{aligned}
\mathbb{E}[\|\epsilon^\top\Phi U\|_{V^{-1}}^2] &= \mathbb{E}[\mathrm{tr}(\Phi U V^{-1}U^\top\Phi^\top)] \\
&= \mathbb{E}[\mathrm{tr}(\Phi^\top\Phi U V^{-1}U^\top)] = \mathrm{tr}(n_o\Sigma_\rho U V^{-1}U^\top) \\
&= n_o\,\mathrm{tr}(UVU^\top U V^{-1}U^\top) \\
&= n_o\,\mathrm{tr}(UU^\top) = n_o\,\mathrm{tr}(U^\top U) = n_o\mathrm{rank}(\Sigma_\rho).
\end{aligned}
$$

Similarly,

$$
\mathbb{E}_\epsilon\left[\|\epsilon^\top\Phi(\mathbf{I}-UU^\top)\}\|_2^2\right] = \mathrm{tr}(\Phi(\mathbf{I}-UU^\top)(\mathbf{I}-UU^\top)\Phi^\top) = \mathrm{tr}(\Phi^\top\Phi(\mathbf{I}-UU^\top)).
$$

Then, by the law of total expectation,

$$
\begin{aligned}
\mathbb{E}\left[\|\epsilon^\top\Phi(\mathbf{I}-UU^\top)\}\|_2^2\right] &= \mathbb{E}[\mathrm{tr}(\Phi^\top\Phi(\mathbf{I}-UU^\top))] \\
&= n_o\,\mathrm{tr}(\Sigma_\rho(\mathbf{I}-UU^\top)) = n_o\,\mathrm{tr}(UVU^\top(\mathbf{I}-UU^\top)) = 0.
\end{aligned}
$$

Combining all things together,

$$
\mathcal{R}_n(\xi;\mathcal{F}) \leq \xi\sqrt{\mathrm{rank}[\Sigma_\rho]/n_o}.
$$

Then, the critical inequality becomes

$$
y\sqrt{\mathrm{rank}(\Sigma_\rho)/n_o} \leq y^2/\beta.
$$

Thus, the critical radius of $\mathcal{F}$ is

$$
\sqrt{\beta\mathrm{rank}(\Sigma_\rho)/n_o}.
$$

$\square$

**Lemma 38** (Variatioanl representation of kernels). *We denote the RKHS associated with a kernel $k(\cdot,\cdot)$ by $\mathcal{H}_k$. Then,*

$$
k(x,x) = \sup_{\{f:\|f\|_k\leq 1, f\in\mathcal{H}_k\}} f^2(x).
$$

*Proof.* We have

$$\sup_{\{f:\|f\|_k \leq 1, f \in \mathcal{H}_k\}} f^2(x) = \sup_{\{f:\|f\|_k \leq 1, f \in \mathcal{H}_k\}} \langle f, k(x, \cdot) \rangle_k^2$$

$$\leq \sup_{\{f:\|f\|_k \leq 1, f \in \mathcal{H}_k\}} \|f\|_k^2 k(x, x) \qquad \text{(CS inequality)}$$

$$= k(x, x).$$

Besides, the equality is satisfied when $f(\cdot) = k(x, \cdot)/\sqrt{k(x, x)}$ noting

$$f^2(x) = k^2(x, x)/k(x, x) = k(x, x), \quad \|f(\cdot)\|_k = \|k(x, \cdot)\|_k/k(x, x) = 1.$$

Thus,

$$k(x, x) = \sup_{\{f:\|f\|_k \leq 1, f \in \mathcal{H}_k\}} f^2(x).$$

$\square$

**Lemma 39** (Relation between $\mathcal{H}_{k_{n_o}}$ and $\mathcal{H}_k$). *We denoting the RKHS associated with a kernel $k(\cdot, \cdot)$ by $\mathcal{H}_k$ and the RKHS with a kernel $k_{n_o}(\cdot, \cdot)$ by $\mathcal{H}_{k_{n_o}}$. Then, we have $\mathcal{H}_k = \mathcal{H}_{k_{n_o}}$. Besides, for $f \in \mathcal{H}_k$, we have*

$$\|f\|_{k_{n_o}}^2 = \|f\|_k^2 + \zeta^{-2} \sum_{i=1}^{n_o} f(x_i)^2.$$

This is stated in [63, Appendix B] without the proof. For completeness, we provide the proof.

*Proof.* We use Mercer's theorem [73, Theorem 12.20]. Then, any element in the RKHS associated with the kernel $k(x, x)$ is represented by

$$f(x) = \sum_{i=1}^{\infty} f_i \psi_i(x).$$

where $\{\psi_i\}_{i=1}^{\infty}$ is an orthonormal basis for $L^2(\rho)$: $\mathbb{E}_{x \sim \rho}[\psi_i(x)\psi_j(x)] = I(i = j)$. Here, we have

$$k(x, x) = \psi^\top(x) \Lambda \psi(x) = \phi^\top(x)\phi(x), \quad \|f\|_k = \tilde{f}^\top \Lambda^{-1} \tilde{f},$$

where $\phi_i(x) = \sqrt{\mu_i}\psi_i(x)$ and $\tilde{f} = \{f_i\}_{i=1}^{\infty} \in \mathbb{R}^{\infty}$.

Then, by letting $\Phi$ be a $n \times d$ matrix s.t. the $i$-th row is $\phi^\top(s_i, a_i)$,

$$k_{n_o}(x, x) = \phi^\top(x)\phi(x) - \phi^\top(x)\Phi^\top(\Phi\Phi^\top + \zeta^2 \mathbf{I})^{-1}\Phi\phi(x)$$

$$= \phi^\top(x)\{\mathbf{I} - \Phi^\top(\Phi\Phi^\top + \zeta^2 \mathbf{I})^{-1}\Phi\}\phi(x)$$

$$= \phi^\top(x)\{\mathbf{I} + \Phi^\top\Phi/\zeta^2\}^{-1}\phi(x) \qquad \text{(Woodbury matrix identity)}$$

$$= \phi^\top(x)(\mathbf{I} + \sum_{i=1}^{n_o} \phi(x_i)\phi(x_i)^\top/\zeta^2)^{-1}\phi(x).$$

Here, let $UVU^\top$ be the eigenvalue decomposition of $\{\Lambda^{-1} + \sum_{i=1}^{n_o} \psi(x_i)\psi(x_i)^\top/\zeta^2\}^{-1} = UVU^\top$. Then,

$$k_{n_o}(x, x) = \psi^\top(x)(\Lambda^{-1} + \sum_{i=1}^{n_o} \psi(x_i)\psi(x_i)^\top/\zeta^2)^{-1}\psi(x)$$

$$= \psi^\top(x)UVU^\top\phi(x)$$

$$= \psi'^\top(x)V\psi'(x). \qquad (U^\top\psi = \psi')$$

Then, any element $f(\cdot)$ in the RKHS associated with the kernel $k_{n_o}(x, x)$ is represented as

$$f(\cdot) = \tilde{g}^\top\psi'(\cdot), \ \tilde{g} \in \mathbb{R}^{\infty},$$

and the associated norm is $\|f\|_{k_{n_o}} = \tilde{g}^\top V^{-1}\tilde{g}$ since $\psi'(\cdot)$ is still an orthnormal basis for $L^2(\rho)$, i.e., $\mathbb{E}_{x \sim \rho}[\phi'_i(x)\phi_j(x)] = I(i = j)$. This immediately implies $\mathcal{H}_k = \mathcal{H}_{k_{n_o}}$.

Finally, we check the relation of the norm:

$$
\begin{aligned}
\|f\|_{k_{n_o}}^2 = \| \sum_{i=1}^{n_o} f_i \psi_i \|_{k_{n_o}} = \|\tilde{f}^\top \psi\|_{k_{n_o}} \qquad\qquad (\tilde{f} = \{f_1, f_2 \cdots\}^\top)\\
= \|\{U^\top \tilde{f}\}^\top U^\top \psi\|_{k_{n_o}}\\
= \|\{U^\top \tilde{f}\}^\top \psi'\|_{k_{n_o}}\\
= \{U^\top \tilde{f}\}^\top V^{-1} U \tilde{f}\\
= \tilde{f}^\top (\Lambda^{-1} + \sum_{i=1}^{n_o} \phi(x_i)\phi(x_i)^\top/\zeta^2)\tilde{f}\\
= \|f\|_k + 1/\zeta^2 \sum_{i=1}^{n_o} \{\tilde{f}^\top \phi(x_i)\}^2 = \|f\|_k + \zeta^{-2} \sum_{i=1}^{n_o} f^2(x_i).
\end{aligned}
$$

$\square$

**Lemma 40.** *Let $\{\hat{\mu}_i\}_{i=1}^{n_o}$ be the eigenvalues of $\mathbf{K}_{n_o}$. Then,*

$$
\sum_{i=1}^{n_o} \frac{\hat{\mu}_i/\zeta^2}{\hat{\mu}_i/\zeta^2 + 1} = \frac{1}{\zeta^2} \sum_{i=1}^{n_o} k_{n_o}(x_i, x_i).
$$

*Proof.*

$$
\begin{aligned}
\sum_{i=1}^{n_o} k_{n_o}(x_i, x_i) &= \sum_{i=1}^{n_o} k(x_i, x_i) - \bar{k}_{n_o}^\top(x_i)\{\mathbf{K}_{n_o} + \zeta^2\mathbf{I}\}^{-1}\bar{k}_{n_o}(x_i)\\
&= \mathrm{tr}\left( \sum_{i=1}^{n_o} k(x_i, x_i) - \bar{k}_{n_o}^\top(x_i)\{\mathbf{K}_{n_o} + \zeta^2\mathbf{I}\}^{-1}\bar{k}_{n_o}(x_i) \right)\\
&= \mathrm{tr}\left( \mathbf{K}_{n_o} \right) - \mathrm{tr}\left( \sum_{i=1}^{n_o} \bar{k}_{n_o}(x_i)\bar{k}_{n_o}^\top(x_i)\{\mathbf{K}_{n_o} + \zeta^2\mathbf{I}\}^{-1} \right)\\
&= \mathrm{tr}\left( \mathbf{K}_{n_o} - \mathbf{K}_{n_o}^2\{\mathbf{K}_{n_o} + \zeta^2\mathbf{I}\}^{-1} \right)\\
&= \mathrm{tr}\left( \{\mathbf{K}_{n_o}^2 + \zeta^2\mathbf{K}_{n_o} - \mathbf{K}_{n_o}^2\}\{\mathbf{K}_{n_o} + \zeta^2\mathbf{I}\}^{-1} \right)\\
&= \mathrm{tr}\left( \zeta^2 \mathbf{K}_{n_o}\{\mathbf{K}_{n_o} + \zeta^2\mathbf{I}\}^{-1} \right) = \sum_{i=1}^{n_o} \frac{\hat{\mu}_i}{\hat{\mu}_i/\zeta^2 + 1}.
\end{aligned}
$$

$\square$

**Lemma 41** (Calculation of localized Rademacher complexity of RKHS balls: Corollary 14.5 in [73]). *Let $\mathcal{F} = \{f \in \mathcal{H}_k : \|f\|_k \le 1\}$ be the unit ball of an RKHS with eigenvalues $\{\mu_j\}_{j=1}^\infty$. Then, the localized population Rademacher complexity is upper-bounded by*

$$
\mathcal{R}_n(\delta; \mathcal{F}) \le \sqrt{\frac{2}{n}}\sqrt{\sum_{j=1}^\infty \min(\mu_j, \delta^2)}.
$$

**Lemma 42** (Upper-bound of expectation of information gains: finite-dimensional models ).

$$
\mathbb{E}[\bar{\mathcal{I}}_{n_o}] \le \mathrm{rank}(\Sigma_\rho)\{\log(1 + n_o/\lambda) + 1\}.
$$

*Proof.*

$$
\begin{aligned}
\mathbb{E}[\bar{\mathcal{I}}_{n_o}] &= \mathbb{E}[\log(\det(\Sigma_{n_o}/\lambda))]\\
&\le \log\det(\mathbb{E}[\Sigma_{n_o}/\lambda]) = \log\det(\mathbf{I} + n_o/\lambda\Sigma_\rho) \qquad \text{(Jensen's inequality)}\\
&\le \mathrm{rank}(\Sigma_\rho)\{\log(1 + n_o/\lambda) + 1\}.
\end{aligned}
$$

The final line is proved as in the proof of theorem 20.

$\square$

**Lemma 43** (Upper-bound of expectation of information gains: RKHS)**.**

$$\mathbb{E}[\mathcal{I}_{n_o}] \leq 2d^*\{\log(1 + n_o/\zeta^2) + 1\}.$$

*Proof.*

$$\begin{aligned}
\mathbb{E}[\mathcal{I}_{n_o}] &= \mathbb{E}[\log(\det(I + \zeta^{-2}\mathbf{K}_{n_o}))] \\
&\leq \sum_{s=1}^{\infty} \log(1 + \zeta^{-2}\mu_s n_o) \qquad\qquad \text{(Refer to [60, Lemma 1])} \\
&\leq \{\log(1 + n_o/\zeta^2) + 1\}2d^*.
\end{aligned}$$

From the second line to the third line, we follow in the proof of theorem 24. $\qquad\square$

# E   Implementation Details

Here we detail all environment details and hyperparameters used for the experiments in the main text.

## E.1   Environment Details

All environments have a maximum horizon length of 500 timesteps. We achieve this by reducing the data collection frequency of the base 1000 horizon environments. We also remove all contact information from the observation and the reward. Finally, to be able to compute the ground truth reward from the state, we add the velocity of the center of mass into the state.

Table 3: Observation and action space dimensions for each of the environments

| Environment | Observation Space Dimension | Action Space Dimension |
|---|---|---|
| Hopper | 12 | 3 |
| Walker2d | 18 | 6 |
| HalfCheetah | 18 | 6 |
| Ant | 29 | 8 |
| Humanoid | 47 | 17 |

Table 4: Ground truth environment reward function used to train the expert and behavior policies as well as evaluate the performance in the learning curves. At time $t$, $\dot{x}_t$ is the velocity of the center of mass in the $x$-axis, $\mathbf{a_t}$ is the action vector, and $z_t$ is the position of the center of mass in the $z$-axis.

| Environment | Ground Truth Reward Function |
|---|---|
| Hopper | $\dot{x}_t - 0.1\|\mathbf{a_t}\|_2^2 - 3.0 \times (z_t - 1.3)^2$ |
| Walker2d | $\dot{x}_t - 0.1\|\mathbf{a_t}\|_2^2 - 3.0 \times (z_t - 0.57)^2$ |
| HalfCheetah | $\dot{x}_t - 0.1\|\mathbf{a_t}\|_2^2$ |
| Ant | $\dot{x}_t - 0.1\|\mathbf{a_t}\|_2^2 - 3.0 \times (z_t - 1.3)^2$ |
| Humanoid | $1.25 \times \dot{x}_t - 0.1\|\mathbf{a_t}\|_2^2 + 5 \times bool(1.0 \leq z_t \leq 2.0)$ |

## E.2   Dynamics Ensemble Architecture and Model Learning

For all of our experiments we use an ensemble of four dynamics models with each model parameterized by a feed-forward neural network with two hidden layers containing 1024 units. The learned model does not predict next state, but instead predicts the normalized difference between the next state and the current state, $s_{t+1} - s_t$. The activation function used at each layer is ReLU. We train all of our ensembles using Adam with learning rate $5 \times 10^{-5}$ and otherwise default hyperparameters. We train each dynamics model for 300 epochs on just the offline dataset for all of our experiments. Please see Table 5 for all values.

Table 5: All hyperparameters used for dynamics model learning

| Hyperparameter | Value |
|---|---|
| Hidden Layers | $(1024, 1024)$ |
| Activation | ReLU |
| Optimizer | Adam |
| Learning Rate | $5 \times 10^{-5}$ |
| Batch Size | 256 |
| Epochs | 300 |

## E.3 Policy Architecture and TRPO Details

We use the open source NPG/TRPO implementation, MJRL [53]. The policy network and the value network are feedforward neural networks with two hidden layers containing 32 and 128 hidden units respectively. Both networks use a `tanh` activation function with the policy network outputting a Gaussian distribution $\mathcal{N}(\mu(s), \sigma^2)$ where $\sigma$ is a trainable parameter. We use Generalized Advantage Estimator (GAE) to estimate the advantages. Please see Table 6 for all values.

Table 6: TRPO/NPG hyperparameter values used in experiments.

| Hyperparameter | Value |
|---|---|
| Policy Hidden Layers | $(32, 32)$ |
| Critic Hidden Layers | $(128, 128)$ |
| Batch Size | 40000 |
| Max KL Divergence | 0.01 |
| Discount $\gamma$ | 0.995 |
| CG Iterations | 25 |
| CG Damping | $1 \times 10^{-5}$ |
| GAE $\lambda$ | 0.97 |
| Critic Update Epochs | 2 |
| Critic Optimizer | Adam |
| Critic Learning Rate | $1 \times 10^{-4}$ |
| Critic L2 Regularization | $1 \times 10^{-4}$ |
| Policy Init Log Std. | -0.25 |
| Policy Min Log Std. | -2.0 |
| BC Regularization $\lambda_{\text{BC}}$ | 0.1 |

## E.4 Hyperparameter Selection

For our core results, we tuned our hyperparameters on a randomly selected seed for `Hopper-v2` and then applied it for all environments. For TRPO, we tuned the conjugate gradient iterations from values 10, 25, and 50; and the conjugate gradient damping coefficients from values 1e-2, 1e-3, 1e-4, and 1e-5. All other hyperparameters were default ones in the MJRL repository [53]. For the BC regularization coefficient we tested values of 0.1, 1e-2, 1e-3, 1e-4, and 1e-5. For the dynamics model architecture we tested 3 different hidden layer sizes: 256, 512, and 1024. Beyond this we used the exact same Adam optimizer and training procedure as [34].

## E.5 Discriminator Update and Cost Function Details

We parameterize our discriminator as a linear function $f(s, a) = w^\top \phi(s, a)$, where $\phi(s, a)$ are Random Fourier Features [51] and $w$ is the vector of parameters for the discriminator. Recall our objective,

$$\min_{\pi \in \Pi} \max_{f \in \mathcal{F}} \left[ \mathbb{E}_{(s,a) \sim d_{\hat{P}}^\pi} (f(s,a) + b(s,a)) - \mathbb{E}_{(s,a) \sim \mathcal{D}_e} [f(s,a)] \right] + \lambda_{\text{BC}} \cdot \mathbb{E}_{(s,a) \sim \mathcal{D}_e} [\ell(a, s, \pi)].$$

Now given a policy $\pi$, we can compute a closed form update for the discriminator parameters $w$ like so

$$\max_{w:\|w\|_2^2 \le \eta} L(w; \pi, \widehat{P}, b, \mathcal{D}_e) := \mathbb{E}_{(s,a)\sim d_{\widehat{P}}^\pi}(f(s,a) + b(s,a)) - \mathbb{E}_{(s,a)\sim \mathcal{D}_e}[f(s,a)]$$

$$\equiv \max_w L_\eta(w; \pi, \widehat{P}, b, \mathcal{D}_e) = \mathbb{E}_{(s,a)\sim d_{\widehat{P}}^\pi}(f(s,a) + b(s,a)) - \mathbb{E}_{(s,a)\sim \mathcal{D}_e}[f(s,a)] - \frac{1}{2} \cdot (\|w\|_2^2 - \eta)$$

$$\Rightarrow \partial_w L_\eta(w; \pi, \widehat{P}, b, \mathcal{D}_e) = \mathbb{E}_{(s,a)\sim d_{\widehat{P}}^\pi}[\phi(s,a)] - \mathbb{E}_{(s,a)\sim \mathcal{D}_e}[\phi(s,a)] - w$$

where $\partial_w L_\eta(w; \pi, \widehat{P}, b, \mathcal{D}_e)$ denotes the partial derivative of $L_\eta(\cdot)$ wrt to $w$. Setting the above expression to 0 and solving for $w$ gives us the closed form solution. Note that even with the BC regularization constraint added into the objective, the solution will still hold.

Now for a given updated $w_t$, we have our cost function $c(s,a) = w_t^\top \phi(s,a) + b(s,a)$ where our penalty, $b(s,a)$, is the maximum discrepancy of our model ensemble predictions. To balance our penalty term with our cost term, we introduce a parameter $\lambda_{\text{penalty}}$ to get the cost

$$c(s,a) = (1 - \lambda_{\text{penalty}}) \cdot w_t^\top \phi(s,a) + \lambda_{\text{penalty}} \cdot b(s,a).$$

In our experiments, $\lambda_{\text{penalty}}$ was the only parameter we varied across environments.

Table 7: $\lambda_{\text{penalty}}$ values used for each environment.

| Environment | $\lambda_{\text{penalty}}$ |
|---|---|
| Hopper | $2.5 \times 10^{-4}$ |
| Walker2d | $1.0 \times 10^{-7}$ |
| HalfCheetah | $1.0 \times 10^{-4}$ |
| Ant | $1.0 \times 10^{-4}$ |
| Humanoid | $5.0 \times 10^{-4}$ |

## F    Additional Experiments

Recall that in our main experiments, we create an extremely small expert dataset containing expert $(s,a)$ pairs by randomly sampling state-action pairs from an expert dataset consisting of state-action pairs from many expert trajectories, and we did that for the purpose of creating an expert dataset where BC almost fails completely. One may wonder what `MILO` would do if we feed `MILO` a complete single expert trajectory. We conduct such experiments in this section. Figure 4 shows the performance of `MILO` with one expert trajectory using the *same hyperparameters* as before. All plots are shown averaged across five seeds. Note that `MILO` is still performs well with one expert trajectory—matching or nearly matching the expert performance across all 5 continuous control tasks.

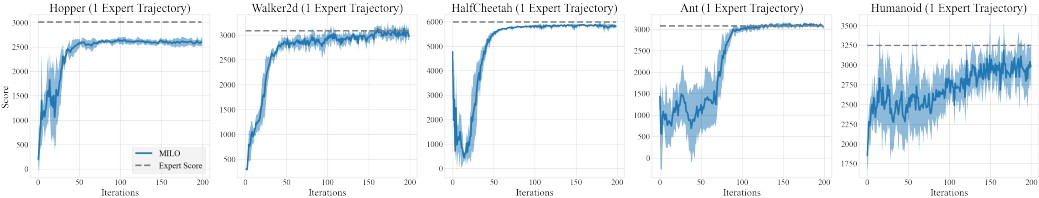

Figure 4: Performance of `MILO` with one expert trajectory. Note `MILO` performance just as well with trajectory inputs as with state-action pair sample inputs.