# OpenReview forum: "Mitigating Covariate Shift in Imitation Learning via Offline Data With Partial Coverage"
_NeurIPS.cc/2021/Conference — NeurIPS 2021 Poster_

### Official Review · Reviewer_WrHs · 2021-07-01

**Rating:** 6
**Confidence:** 3

**Summary:**

This work proposes a new method for offline imitation learning. The setup is slightly different than usual as it supposes that, additionally to the expert dataset we want to imitate, we have access to a large dataset of non-expert interactions.
The method is the following:

1) learn a model of the transition kernel from the data.

2) Along with the model, learn an “uncertainty penalty” (the converse of an exploration bonus) that will encourage the agent to stay on the expert support.

3) Train *offline* a min-max model-based policy-optimization with the uncertainty penalty.

The contributions also include theoretical analysis on the proposed method in simplified cases.

**Limitations And Societal Impact:**

No direct societal impact.

**Main Review:**

Thank you for this interesting work. A few questions, particularly on the experimental part:

Methods:

- Why don’t you also use the expert dataset for training the model? I see no interest in leaving it out.

- Have you thought about any connection to the GAIL algorithm? What you call the pessimistic model-based offline IL is, I am not mistaken, an instantiation of the “adversarial imitation learning” framework with trajectories played in the model rather than online, and an additional non-exploration bonus.

Experiments:

- It is said (both in title and analysis) that the method does not require full coverage of the state space. I would have appreciated to see an experiment with the results of the methods when decreasing the size of $D_o$ (which is really important: 1M samples is not quite a “partial coverage” for these locomotion environments).

- I find it slightly problematic that no presented baselines make use in a principled way of $D_o$ (BC just learns from it like it was expert data). In particular, one natural baseline (feel free to disagree) to me would be to train any offline RL learner on $D_e+D_o$ with the following reward: 1 for any sample in $D_e$, 0 for any sample in $D_o$ (in the spirit of SQIL [1]). In a GAIL-like manner, you could also define the reward as $log(D)$ with D a discriminator between $D_e$ and $D_o$ and train an offline agent on  $D_e+D_o$ with this reward function.

- It would be great to detail in the appendix how hyperparameter selection was performed. In particular, I would encourage you to check out [2], that encourages not to select hyperparameters on the underlying reward function when performing imitation.

- Can you clarify Appendix E.1? Why did you modify the base environments most people in the community use? I find this quite surprising and I do not feel comfortable with this. According to me, you should either switch back to the widely used environments or justify very clearly why you needed these modifications.

- From Fig.2 it looks like the method is initialized with BC. As far as I can tell, this is not specified in the paper. Otherwise, why is MILO’s curve not starting at random behavior’s performance?


I would increase my score if you can provide clear explanations concerning the experimental setup. Bonus if you can elaborate a bit on the link to GAIL and add these additional baselines.



[1] Reddy, S. et. al., 2019. Sqil: Imitation learning via reinforcement learning with sparse rewards. arXiv preprint arXiv:1905.11108.
[2] Hussenot, Leonard, et al. "Hyperparameter Selection for Imitation Learning." arXiv preprint arXiv:2105.12034 (2021).



# Update after rebuttal

I would like to thank the authors for clearly answering my questions and providing additional experiments/ablation supporting the method's results.

I encourage the authors to include the feedback they provided here in the paper itself.

I would say that the methodology for hyperparameter selection is not following best practices (e.g. taking best of with/without BC init) and I encourage the authors to improve this for future works. Yet, most of my concerns were answered and, although HP selection could be improved, the presented results look valid.

I will thus increase my score to 6.

**Time Spent Reviewing:**

4

---

> ### Author Response · Authors · 2021-08-10
> **Response to Reviewer Feedback**
>
> Thank you for your kind comments and thorough review.
>
> ### Using Expert Data in Dynamics Training
> In our setting, we did not assume that we had (s, a, s’) triple style information from the expert. Instead, following the BC / GAIL setting and for a fair comparison to BC, we only assume we have access to (s,a) style expert datasets. If we have (s, a, s’) expert samples, then as you pointed out we can definitely use them in our dynamics training which will make the combined dataset to cover the expert distribution automatically (i.e., the partial coverage condition will be automatically met).
>
> ### Connections to GAIL
> You are right in that our Pessimistic Min-Max procedure does fall under the broader umbrella of adversarial imitation learning methods but has a few important distinctions. One of our main contributions is the addition of a penalty to the cost which we show both in theory and empirically is critical in mitigating covariate shift in offline setting (i.e., unlike GAIL, we do not have online interaction). Furthermore, different from GAIL, we make use of Integral probability metric (IPM) which leads to minimizing the Maximum Mean Discrepancy (MMD) rather than the Jenson-Shannon distance.
>
> ### Experimentation on Coverage with Less Data
> Thank you for the suggestion and we will definitely consider strengthening our coverage ablation from this angle. We would like to clarify that reducing the size of the offline dataset affects the statistical error (i.e. the $1/\sqrt{N}$ term in Section 4) but not the coverage term. The coverage term is related to the density ratio [Line 187 on Page 5]. This is the population quantity, which does not depend on the sample size. For example, if $\rho$ has zero support on some state-actions visited by the expert, regardless of how many offline samples we may have from $\rho$, the offline data still cannot cover states visited by the expert .
>
> We would like to point out that we measured decreased coverage of our offline dataset by reducing the behavior policy quality used to collect the dataset. This strategy is widely used in the Offline RL literature [33], [72] including the most popular D4RL benchmark datasets which maintain the same offline dataset size across datasets with different coverage. For example, an offline dataset of 1M samples from a randomly initialized behavior policy would most likely not have good coverage over the expert state-action distribution. We show the described effects in Table 2 Section 6.2 of our experiments where we decrease the coverage of our offline datasets and see a deterioration in MILO’s performance. Note that even for a state-of-the-art offline RL algorithm like CQL that has access to true reward signals, the algorithm’s performance deteriorates with such a random offline dataset with 1M samples [37].
>
> ### Comments on Baselines and Experiment Suggestions
>
> **SQIL type experiment suggestion**: Thank you for this suggestion. We have run some preliminary experiments following your suggestions. With the SQIL reward parameterization, we tried running both in a model-free and model-based version: the model-free version uses SAC with the described SQIL-like replay buffer, and the model-based version uses MOPO [78] as the offline learner. We will add the plot comparisons across all datasets in the final version but for now here are some preliminary results across 3 seeds for HalfCheetah (the environment MOPO performed most favorably on): both methodologies peaked at an average score of ~800 and dropped to random performance by 500 iterations. One preliminary observation we see is that similar to the online setting for SQIL, we generally see a deterioration in performance when run long enough.
>
> One potential issue of the this reward and replay buffer design is that in the extreme ideal case where $D_o$ and $D_e$ are both from the expert (i.e., perfect coverage with density ratio = 1, but the learner is unaware of this fact), such reward design will cause conflict: we assign reward 1 to samples in $D_e$ but reward zero to samples in $D_o$, despite both datasets being sampled from the same distribution!  We do not immediately see how to rigorously prove the optimality of the procedure.
>
> **GAIL experiment suggestion**:  Due to limited time, we were unable to run your suggested experiment using a GAIL-like reward function for this rebuttal but will continue to investigate. We want to bring to the reviewer’s attention that our ablation study without the reward penalty is essentially a model-based GAIL procedure but with a discriminator using an RBF kernel. One potential issue of this approach is that again in the extreme ideal case where $D_o$ and $D_e$ are both sampled from the expert, the divergence will be zero under any discriminator, i.e., no discriminator will be able to distinguish these two datasets. Thus, it could potentially output an arbitrary discriminator which potentially will hurt the policy optimization.
>
> **Current Baselines**: We would like to emphasize that our baseline using Offline ValueDICE [36] is a fundamentally valid and recently developed competitive algorithm for handling offline and off-policy data. ValueDICE leverages the DICE technique ([43,78]) which is a principled way of learning from off-policy data. Moreover, the offline RL version of ValueDICE has been investigated by the authors themselves (Appendix C [36]).
>
> ### Hyperparameter Selection
> Thank you for pointing this out. For our core results, we tuned our hyperparameters on a randomly selected seed for Hopper and then applied it for all environments. For TRPO, we tuned the conjugate gradient iterations (tried [10, 25, 50]) and conjugate gradient damping coefficients ([1e-2, 1e-3, 1e-4, 1e-5]). All other hyperparameters were default ones in the MJRL repository [49]. For the BC regularization coefficient we tried [0.1, 1e-2, 1e-3, 1e-4, 1e-5] and for the dynamics model architecture we tested 3 different hidden layer sizes [256, 512, 1024]. Beyond this we used the exact same Adam optimizer and training procedure as [33]. We will add these details.
>
> ### Environment Modifications
> We follow the practices of the state-of-the-art Model-based RL literature for MuJoCo environments (e.g. SLBO [Luo et al], Model-Ensemble TRPO [Kurutach et al], MOReL [Kidambi et al]). Note that we build our framework on top of MOReL’s codebase and continue to use their environment definitions with guidance from details provided by a model-based RL benchmarking paper [Wang et al.]. Please see Appendix F from [Wang et al] for a detailed discussion about our environment design choice.
>
> ### Warm-started/Initialized with BC
>  Yes, our core results are initialized with 1 epoch of BC on the offline dataset + expert dataset. In order for fair comparison, we did this for all of our benchmarks as well and reported the best performance between (with/without BC warmup). Thank you for pointing this out and we will clarify this in the final draft.
>
> ### References
> [Luo et al] Algorithmic Framework for Model-Based Deep Reinforcement Learning with Theoretical Guarantees
>
> [Kurutach et al] Model Ensemble Trust-Region Policy Optimization
>
> [Kidambi et al] MOReL: Model-Based Offline Reinforcement Learning
>
> [Wang et al] Benchmarking Model-Based Reinforcement Learning

---

> ### Author Response · Authors · 2021-08-13
> **Update: Experiment Suggestion - GAIL style cost Log(D(s,a))**
>
> As an update to our original response, we trained a GAIL-like discriminator on $D_o$ and $D_e$ until convergence and then populated the replay buffer with the learned rewards. We then trained a policy with SAC using this replay buffer where each batch of samples from the replay buffer was half offline data samples and half expert samples. We tried this with varying expert dataset sizes (125, 1000, 5000) and found that across 3 seeds none of the learned policies exceeded 10% of the expert performance on Hopper despite having more expert samples than MILO.

---

### Official Review · Reviewer_UUFL · 2021-07-15

**Rating:** 6
**Confidence:** 4

**Summary:**

The authors present MILO an offline imitation learning algorithm that uses a learned model of the environment to generate trajectories from arbitrary policies. This allows to reuse similar algorithms than in the online case such as GAIL or Wasserstein GAIL.  In addition, the authors use a bonus (pessimism) to penalize data that is generated in places with high model uncertainty. This can be achieved using an ensemble. They present experimental results as well as an ablation study on OpenAI Gym.

**Limitations And Societal Impact:**

Yes the authors address the limitations and potential negative societal impact of their work adequately.

**Main Review:**

I think the paper is still not ready for publication in a venue such as Neurips for the following reasons.

First, the literature may have not been studied properly. Indeed,  the idea of offline imitation or offline inverse reinforcement learning using a set of expert trajectories and a set of non-expert trajectories to add dynamics information is not new :

 - A Cascaded Supervised Learning Approach to Inverse Reinforcement Learning (ECML 2013)

Unfortunately, this paper is not cited.  Also the authors do not cite DART (DART: Noise Injection for Robust Imitation Learning) which is also one way to avoid covariate shift by adding noise when collecting trajectories from the expert.  Finally the idea of framing the IL/IRL problem as a min-max problem dates at least from the 2004 seminal paper of Peter Abbeel (Apprenticeship Learning via Inverse Reinforcement Learning) and could have been cited.

Second, the experiments section should prove the main point against online methods raised by the authors in the introduction: "online interactions are often costly and prohibitive for real world applications where active trial-and-error exploration in the environment could be unsafe or impossible". Unfortunately, I was not convinced. Indeed, in the experiments section the non-expert data set to train the generative model uses 1 million transitions. Therefore, one possible experiment could be to see how much data an online imitation learning algorithm such as GAIL needs to generate to reach the same performance as MILO and show that MILO is at least one order of magnitude below that. Without this experiment we can't really judge how data-efficient is MILO compared to GAIL. One other interesting experiment could be to see what is the proportion of low-score trajectories generated by GAIL compared to the dataset used to train the generative model. This experiment would tell you how unsafe GAIL is compared to MILO. Those two experiments could prove clearly the point you were making in the introduction. I encourage strongly the authors to run those experiments which will make the paper stronger.

Third, you should have an offline IRL or IL baseline (such as CSI) and not baselines that are artificially weakened such as behaviour cloning baselines with sparse expert data. It is known that this is an adversarial setting for these methods. The same goes for MILO when the non-expert data is random which results on poor results as shown in Table 2.

Fourth, the analysis section could be moved in the appendix section because it covers cases such as discrete MDPs and GPs that are not related to the practical algorithm presented and used in the experiments. Indeed different instanciations of the general algorithms are presented and this can be confusing for the reader. For instance the algorithm is presented with and without behaviour cloning loss and then the exact cost function is only provided in the appendix. An exact and detailed description of the algorithm used for the experimental section should be done in the main paper.  In addition, the authors should focus more on how could we learn the model \hat{P} on more challenging environments. This is the main bottleneck of this algorithm. This scaling problem seems quite challenging if the states are images or high-dimensional in general. How much data will be needed in that case? Should this model be learned at the latent level?

Overall I think the paper has some strong points like the idea of learning the bonus b with an ensemble of networks and could be published in a top-tier conference. However, there is still a lot of work on the experiment section to show quantitatively what is the advantage of using offline methods versus online methods and on the method section to present more clearly the exact algorithm.


**Time Spent Reviewing:**

6

---

> ### Author Response · Authors · 2021-08-10
> **Response to Reviewer Feedback**
>
> Thank you for your time and thorough review.
>
> ### Missing Citations
> Thank you for these suggestions.  We will definitely cite these papers. That said, MILO presents a solid contribution beyond these works. This is because we use penalty based bonuses and formally show that we can mitigate the covariate shift problem (i.e., $H^2$ versus $H$). Following DAgger which rigorously shows one can improve from $H^2$ to $H$ via interactive experts, our work rigorously shows that one can improve from $H^2$ to $H$ via offline data with partial coverage.
>
> **CSI**: Thank you - we will cite this work in our final draft. Indeed CSI tackles IRL in an offline manner, using a static offline dataset along with the expert dataset to learn a reward function that is then used in LSPI. We would like to note that algorithms like LSPI --- even with true reward signals, require full coverage in offline data (e.g., see the global coverage assumption in the seminar work FQI [42], and more recent work from [19] which specifically studies LSPE), while we are focus on the setting where the offline data does not have global coverage which is why we have to leverage pessimism inside the learning process.  Essentially, CSI cannot mitigate covariate shift under the partial coverage setting.
>
> **DART**: We agree with the reviewer that DART does tackle covariate shift and will cite this in the final draft. We emphasize, though, that the setting is different. MILO works in a purely static, offline setting. On the other hand, DART iteratively queries the expert and requires actively collecting expert demonstrations every time the algorithm updates the noise parameters.
>
> **Apprenticeship Learning**: We agree with the reviewer that Abbeel & Ng’s “Apprenticeship Learning for Inverse Reinforcement Learning” is a seminal work in this line of research and will incorporate the citation in the final draft. We would like to point out that their approach will suffer ambiguity as pointed out by MaxEnt-IRL [Ziebart et al], i.e. a reward function predicting zero is an optimal solution. However, our min-max objective with min over policies first and then max over discriminators avoids such trivial solutions.
>
> ### Comparison Against Online Algorithms
>  We agree that comparisons against GAIL would be interesting and we will investigate this in the final draft. We would like to highlight that the benefits of training sequential decision making agents in an offline manner and the resulting improvement of not needing online interaction is a widely accepted stance. In Levine et al.’s survey on Offline Reinforcement Learning, we quote
> >“the fact that [RL] algorithms provide a[n] … online learning paradigm is … one of the biggest obstacles to … widespread adoption. In many settings, this sort of online interaction is impractical either because data collection is expensive … and[/or] dangerous.”
>
> One could imagine practical applications where we may have access to a dataset of interactions but not the ability to actively deploy a suboptimal policy for online interactions like GAIL would require.
>
> We ran GAIL with 10 expert trajectories (a much larger expert dataset) for 5 seeds for both Hopper as well as Walker2d. After 1M online samples GAIL achieved 83% and 50% of the expert performance respectively on Hopper and Walker2d -- worse than MILO. We will report more thorough comparisons in our final draft.
>
> ### Move Portions of Analysis to Appendix
> We thank the reviewer for the suggestion, but we would like to respectfully disagree with this change. We want to emphasize that our PAC analysis is an important contribution. Note that Reviewer 1 commented “the paper is rigorously defined mathematical terms” and Reviewer 2 commented “Theory on non-parametric models in offline RL is new”. Especially, our theory is not just the combination of the existing theory of offline RL and imitation learning. The analysis for non-parametric models is totally novel even if we put it under the literature of theoretical offline RL.
>
> Though we admit there is still some gap between the theory and empirics, compared to most of the existing RL papers only focusing on one viewpoint, our paper is trying to fill this gap by providing a detailed discussion from both viewpoints. We strongly believe that rigorously showing the correctness with respect to sample efficiency and provable global optimality in simplified settings like tabular MDPs and linear/kernel function approximation is important and useful in guiding the design of our algorithm to more complex settings.
>
> ### Scaling MILO up
> We agree that scaling MILO to more complex environments is an interesting direction for future work. First we emphasize that MILO is a modular algorithmic framework that allows for advancements made in the Model-Based RL literature to be applied. As the reviewer pointed out, one interesting direction for higher dimensional states such as images may be to learn a model like Dreamer-v2 [Hafner et al] in a latent space and design the reward penalty based on latent disagreement. Next, we note that not every control problem needs to be framed as pixel-to-torque control, and model-based control with compact state representation is widely used in the literature. We emphasize that we already expanded the scope of environments experimented on the MuJoCo benchmark suite from the Offline RL literature: SOTA offline rl algorithms such as MOPO [76], MOREL[33], COMBO[75], CQL [37] experiment up until Ant and do not include Humanoid.
>
> ### Artificially Weakened Baselines
> Thank you for suggestions for adding more baselines such as CSI. We will look into incorporating more in our final draft. We would like to emphasize that our baseline using Offline ValueDICE [36] is a fundamentally valid and recently developed competitive algorithm for handling offline and off-policy data. ValueDICE leverages the DICE technique ([43,78]) which is a principled way of learning from off-policy data. Moreover, the offline RL version of ValueDICE has been investigated by the authors themselves (Appendix C [36]).
>
> ### References
> [ Ziebart et al ] Maximum Entropy Inverse Reinforcement Learning
>
> [Levine et al.] Offline Reinforcement Learning: Tutorial, review and perspective on open problems
>
> [Hafner et al.] Mastering Atari with Discrete World Models

---

> > ### Comment · Reviewer_UUFL · 2021-08-13
> > **Response to authors.**
> >
> > The authors have answered my questions and are making a clear effort to improve the paper and its contributions.
> > I will therefore increase my score from 4 to 6.

---

> ### Author Response · Authors · 2021-08-13
> **Update: More detailed Comparison to CSI**
>
> We would like to point out 4 additional key differences between CSI and MILO:
>
> 1. Coverage: as Corollary 1 in CSI indicates, to learn a meaningful reward, CSI requires the expert distribution to have full coverage, i.e., $\rho_E(s) > 0$ for all $s$; MILO doesn’t require such full coverage assumptions.
> 2. CSI focuses on learning rewards, and does not discuss how to perform policy optimization to find a policy that imitates the expert policy; MILO focuses on how to perform policy optimization under expert data and offline data that does not have full coverage on the state space.
> 3. Covariate shift: Note that the BC policy $\pi_C$ is indeed the optimal policy with respect to the reward $\hat{r}(s,a)$. Thus running RL under $\hat{r}(s,a)$ at best learns the BC policy $\pi_C$, which suffers from covariate shift. On the other hand, MILO is explicitly trying to combat covariate shift.
> 4. CSI requires a deterministic, optimal expert. MILO does not require such assumptions on the expert policy.
>
> We will cite and include a discussion in the final version.

---

### Official Review · Reviewer_aRRv · 2021-07-16

**Rating:** 7
**Confidence:** 4

**Summary:**

This paper studies imitation learning without access to the environment. The agent is provided with two datasets: one containing expert policy state-action pairs and one containing state-action-next state triplets collected from the environment. The authors propose the algorithm MILO that leverages pessimism to mitigate covariate shift and provide performance upper bound in discrete MDPs, kernelized nonlinear regulators, and Gaussian processes. The authors show improved performance achieved by MILO compared to other baselines such as behavioral cloning in MuJoCo control tasks.

**Limitations And Societal Impact:**

The authors discuss the limitations and potential negative societal impacts.

**Main Review:**

Originality:
- Theory on non-parametric models in offline RL is new.
- The authors derive bounds depending on the relative condition number w.r.t. expert policy in the KNR setting. In linear function approximation, the presented bound depends on the rank of offline data instead of assuming a full rank covariance.
- The authors thoroughly discuss related work.
- Novelty in algorithmic design is limited (e.g. can be considered a natural variant of MOPO in IL utilizing a cost function class).

Quality:
- The claims are supported with both theoretical analysis and empirical studies.
- BC can perform much better than MILO in certain scenarios and it is unclear how one should choose between these two algorithms based on the datasets. For example, when the expert dataset has good coverage but the offline dataset doesn’t, MILO performs worse than BC by avoiding parts of the environment that are not covered in the offline dataset. On the other hand, when the expert dataset is extremely small but the offline dataset is large with good coverage on expert policy, MILO performs well (e.g. the setting considered in the experiments).

Clarity:
- The paper is very well-written and easy to follow.

Significance:
- The message that pessimism only requires partial coverage is already discussed in prior works.

Questions/comments:
- Below Theorem 3, the authors compare MILO upper bound $O(H/\sqrt{N})$ with BC upper bound $O(H^2/\sqrt{N})$ in [3]. However, the upper bound of BC in [3] is not tight and the comparison is not fair. A tight upper bound of BC is $O(H^2/N)$ given in [47] which is much faster.
- Could you explain the particular choice for characterization of expert policy coverage, e.g. using a ratio in discrete MDP setting? Why didn’t you use alternative measures such as the ones mentioned in Remark 26 in Appendix C.3?

Minor points:
- $b(s,a)$ is referred to as penalty and bonus interchangeably.
- Should cost function class in line 103 be $\mathcal{F} \in \mathcal{S} \times \mathcal{A} \rightarrow [0,1]$?
- A few capitalization errors in the references.

___
After the rebuttal:

I thank the authors for responding to my questions. I am happy with the answers and stick to my score in favor of accepting this paper.

**Time Spent Reviewing:**

5

---

> ### Author Response · Authors · 2021-08-10
> **Response to Reviewer Feedback**
>
> Thank you for the thorough review and suggestions for improvement.
>
> ### BC Upper Bound in [3] is not Tight
> We will mention the relation in the final draft. The reason we compare the upper bound in [3] to our bound is that [3] considers general function approximation settings like us and also considers general stochastic expert policy. The algorithm and result in [47] are specialized to the tabular setting, and the H^2/N rate for general stochastic experts only holds in expectation, rather than as a high probability statement. To the best of our knowledge, it is unclear how to (1) extend their algorithm/analysis to function approximation setup, and (2) whether their current analysis for stochastic experts can be modified for a high probability statement.
>
> ### Particular Choice for Characterization of Expert Policy Coverage
> The reason we use the density ratio is that standard offline RL literature (with PAC guarantee) such as [4], [15],[40] use this assumption to capture the concept of the coverage. Such density ratio based coverage concept is also used in online RL as well, e.g., Conservative Policy Iteration [Langford & Kakade]. We also demonstrate with linear function and kernel, we can refine the density ratio to a relative condition number which previously was used as a coverage condition for online Natural Policy Gradient analysis [Agarwal et al 20].
>
> ### Minor Points
> Thank you for catching typos. We will fix these.
>
> ### References
> [Agarwal et al] On the Theory of Policy Gradient Methods: Optimality, Approximation, and Distribution Shift, JMLR.

---

### Official Review · Reviewer_Fuia · 2021-07-20

**Rating:** 7
**Confidence:** 3

**Summary:**

This work considers an offline imitation learning setting where both expert and sub-optimal demonstrations are assumed to be known. While the covariate shift issue (the gap between training and test distributions) is a critical problem in offline imitation learning, e.g., behavioral cloning, authors address this issue by training a dynamics model through the sub-optimal demonstration (having state-action-successor state triples) and applying pessimistic penalty (incentivizing visiting the state-action pairs in the expert demonstration). With the theoretical derivations for various settings for dynamics modeling and the assumption that the demonstration does not have to cover the entire state-action space, authors show the theoretical guarantee of their proposed method. Also, the empirical results support their claim by showing that MILO (proposed algorithm) outperforms ValueDICE and BC.

**Limitations And Societal Impact:**

The authors pointed out the negative biases that are inherently given in expert demonstration, which I believe hasn’t been addressed in the submission. I couldn’t clearly imagine how to perfectly solve this but I believe recent works on using ranking in expert demonstrations might be helpful.

**Main Review:**

I’m generally satisfied with the submission due to the authors’ clear writing, various experiments (including ablation studies like the impact of coverage), and rigorously defined mathematical terms. While I believe the submission is sufficient to be accepted, there are minor comments below:
- I wonder why the authors did not check the performance of other recent baselines on offline imitation learning that cite the ValueDICE paper. For example,
  - Jarret et al., “Strictly batch imitation learning by energy-based distribution matching,” (NeurIPS 2020, the reference [28] in the submission)
  - Chan et al., “Scalable Bayesian inverse reinforcement learning,” (ICLR 2021, the reference [13] in the submission)
- The citation style is not consistent with the normal NeurIPS citation style.
- Would you please increase the size of figures and fonts in Figures 2, 3, and Table 1,2?


**Time Spent Reviewing:**

5

---

> ### Author Response · Authors · 2021-08-10
> **Response to Reviewer Feedback**
>
> Thank you for your review and your suggestions for improvement.
>
>
> ### Comparisons with Other Baselines
> Both the papers mentioned differ from MILO in terms of the problem setting. These papers present methods that indeed learn in an ‘offline’ setting; however, neither makes use of an additional static dataset in their setup like MILO does. Rather, both papers use generative modeling where they augment expert data with samples from a model learned from the expert data. We also want to note, as investigated by the authors in [48], the imitation learning setting used by the other works [13,28] cannot avoid covariate shift information theoretically in the worst case, indicating that additional assumptions/help was needed. Our work aims to investigate how much benefit one can leverage upon from an additional static offline dataset in order to mitigate covariate shift.
>
>
> ### Citation Style / Figure Formatting
> Thank you for pointing out the citation style inconsistencies and suggestions for the figures. We will fix this in the final draft.
>
>
> ### Incorporating Ranking
> Recent works on using ranking in expert demonstrations might be helpful: Thank you for the suggestion. We will investigate this literature.

---

> ### Comment · Reviewer_Fuia · 2021-08-22
> **I will maintain my score.**
>
> I read the authors' responses and other reviewers' comments. I'm satisfied with the responses and will maintain my score.

---

### Decision · Program_Chairs · 2021-09-27

**Decision:**

Accept (Poster)

**Comment:**

The reviewers in most part agree that paper is well-written, is tackling an interesting problem and also the authors have engaged constructively during the reviewer feedback period and addressed the main concerns and questions. Please address the minor points raised by the reviewers in the final version.